# H3K9 methylation extends across natural boundaries of heterochromatin in the absence of an HP1 protein

Rieka Stunnenberg[1,2], Raghavendran Kulasegaran-Shylini[1,†], Claudia Keller[1,2,‡], Moritz A Kirschmann[1,§], Laurent Gelman[1] & Marc Bühler[1,2,*]

## Abstract

Proteins of the conserved HP1 family are elementary components of heterochromatin and are generally assumed to play a central role in the creation of a rigid, densely packed heterochromatic network that is inaccessible to the transcription machinery. Here, we demonstrate that the fission yeast HP1 protein Swi6 exists as a single highly dynamic population that rapidly exchanges *in cis* and *in trans* between different heterochromatic regions. Binding to methylated H3K9 or to heterochromatic RNA decelerates Swi6 mobility. We further show that Swi6 is largely dispensable to the maintenance of heterochromatin domains. In the absence of Swi6, H3K9 methylation levels are maintained by a mechanism that depends on polymeric self-association properties of Tas3, a subunit of the RNA-induced transcriptional silencing complex. Our results disclose a surprising role for Swi6 dimerization in demarcating constitutive heterochromatin from neighboring euchromatin. Thus, rather than promoting maintenance and spreading of heterochromatin, Swi6 appears to limit these processes and appropriately confine heterochromatin.

**Keywords** H3K9 methylation; heterochromatin; HP1 dynamics; Swi6; Tas3
**Subject Categories** Chromatin, Epigenetics, Genomics & Functional Genomics; Transcription
**The EMBO Journal (2015) 34: 2789–2803**

## Introduction

Eukaryotic genomes are packaged into a complex structure known as chromatin. The basic unit of chromatin is the nucleosome, which consists of two copies each of the histone proteins H2A, H2B, H3, and H4. The flexible N-termini of histone proteins are subject to various posttranslational modifications associated with different types of chromatin. Originally defined cytologically as chromosome regions that do not undergo post-mitotic decondensation but remain condensed during interphase, a distinct type of chromatin referred to as heterochromatin is generally characterized by histone hypoacetylation and specific methylation of lysine 9 of the histone H3 tail (H3K9me). This mark is a binding site for proteins containing a so-called chromodomain (CD), such as proteins of the heterochromatin protein 1 (HP1) family that recognize and bind methylated H3K9 via their CDs (Eissenberg & Elgin, 2000; Bannister *et al*, 2001; Lachner *et al*, 2001).

HP1 proteins have long been thought to play a central role in the creation of a rigid, densely packed heterochromatic network that is inaccessible to the transcription machinery. However, several observations in the last decade have challenged this view (Buhler & Moazed, 2007; Grewal & Elgin, 2007). There have been numerous reports of low-level transcription of heterochromatic regions in various organisms (Rouleux-Bonnin *et al*, 1996, 2004; Lorite *et al*, 2002; Azzalin *et al*, 2007; Li *et al*, 2008; Pezer & Ugarkovic, 2008), and *Schizosaccharomyces pombe* mutants have been identified in which heterochromatic reporter genes are expressed without notable differences in H3K9 methylation or HP1 association (Buhler *et al*, 2007; Keller *et al*, 2012). Furthermore, genomewide studies of chromatin accessibility using *in vivo* methylation by the DNA adenine methyltransferase (Dam) in *C. elegans* showed only little variation (Sha *et al*, 2010), and DNA-binding factors seem to have normal access to repressed sites, even in highly compacted mitotic chromosomes (Verschure *et al*, 2003; Chen *et al*, 2005). Finally, fluorescence recovery after photobleaching (FRAP) experiments in mammalian and *S. pombe* cells revealed that HP1 proteins are mobile molecules (Cheutin *et al*, 2003, 2004; Festenstein *et al*, 2003). Previous kinetic modeling indicated the existence of at least two kinetically distinct populations of the *S. pombe* HP1 protein Swi6 in heterochromatin, arguing for a stochastic model of heterochromatin in which Swi6 acts not solely by forming static

1  Friedrich Miescher Institute for Biomedical Research, Basel, Switzerland
2  University of Basel, Basel, Switzerland
   *Corresponding author. Tel: +41 61 696 04 38; E-mail: marc.buehler@fmi.ch
   †Present address: Wellcome Trust Centre for Cell Biology, Institute of Cell Biology, School of Biological Sciences, The University of Edinburgh, Edinburgh, UK
   ‡Present address: Max Planck Institute of Immunobiology and Epigenetics, Freiburg, Germany
   §Present address: Center for Microscopy and Image Analysis, University of Zurich, Zurich, Switzerland
   [The copyright line of this article was changed on 8 December 2015 after original online publication.]

oligomeric networks. These studies were performed in *swi6Δ* cells in which GFP-Swi6 was ectopically expressed under the control of a thiamine-repressible promoter (Cheutin *et al*, 2004). Whether Swi6 kinetics differ when Swi6 is expressed from its endogenous locus is not known.

In *S. pombe*, transcriptional activity within heterochromatin is particularly evident at pericentromeric repeat sequences, where RNA polymerase II is essential for RNA-interference (RNAi)-mediated assembly and silencing of heterochromatin (Volpe *et al*, 2002; Djupedal *et al*, 2005; Kato *et al*, 2005). Moreover, siRNAs homologous to centromeric sequences accumulate to relatively high levels (Reinhart & Bartel, 2002; Buhler *et al*, 2008; Halic & Moazed, 2010). A key factor for RNAi-mediated deposition of H3K9me2 is the RNA-induced transcriptional silencing (RITS) complex. RITS is composed of Tas3, the CD containing protein Chp1, and the siRNA-binding protein Ago1 (Verdel *et al*, 2004). Together with long chromatin-associated transcripts, RITS acts as a hub for the assembly of machineries that amplify heterochromatic siRNAs, modify histones, and silence gene expression (Holoch & Moazed, 2015). Besides centromeres, constitutive heterochromatin is also found at telomeres and the silent mating type locus in *S. pombe*. However, in contrast to centromeric heterochromatin, the RNAi machinery is dispensable to the maintenance of the repressed state at these regions (Jia *et al*, 2004; Kanoh *et al*, 2005). Instead, the non-canonical polyA-polymerase Cid14 and Swi6 are essential to the elimination of heterochromatic RNA transcripts (Buhler *et al*, 2007; Keller *et al*, 2010). We have recently demonstrated that Swi6 efficiently captures heterochromatic RNAs, which thereby become translationally incompetent and are eventually degraded. Importantly, RNA binding to the hinge region of Swi6 competes with methylated H3K9 for binding to the CD, predicting that Swi6 dynamics *in vivo* might be influenced by RNA (Keller *et al*, 2012).

The requirement for transcription in heterochromatin assembly seems difficult to reconcile with the silencing of heterochromatic genes. One possible solution to this paradox is the need for a pioneering round of transcription to initiate heterochromatin assembly. It has been suggested that Swi6 dissociates from chromatin due to H3S10 phosphorylation during M and S phase, opening up a window of opportunity for the transcription of centromeric repeats. Once assembled, transcription would be silenced by heterochromatin throughout the remainder of the cell cycle (Kloc & Martienssen, 2008). Consistent with this, RNA polymerase II occupancy on centromeric repeats is highest in S phase and transcripts corresponding to centromeric repeat sequences accumulate (Chen *et al*, 2008; Kloc *et al*, 2008). Alternatively, transcription could also occur outside S phase, with the resulting RNA immediately degraded co-transcriptionally (Buhler, 2009; Castel & Martienssen, 2013). However, such highly unstable RNA is difficult to demonstrate biochemically. Thus, although pericentromeric heterochromatin becomes more permissive to transcription in S phase (Chen *et al*, 2008), the actual transcriptional status of heterochromatin, particularly in the G2 phase of the *S. pombe* cell cycle when H3K9me2 levels are the highest, has remained elusive due to technical limitations.

Here, we demonstrate that Swi6 exists as a single dynamic population in constitutive heterochromatin when expressed from its endogenous locus, with little if any Swi6 molecules persistently bound to heterochromatin. Swi6 rapidly exchanges *in cis* and *in trans* between different heterochromatic regions and is not obligatory for the maintenance and spreading of H3K9 methylation. Instead, our results disclose an unexpected activity of Swi6 in demarcating constitutive heterochromatin from neighboring euchromatin. Our results support a model in which Swi6 dimer formation is important in counteracting the spread of H3K9 methylation, which is mediated by a mechanism that requires polymeric self-association properties of the Tas3 subunit of the RITS complex.

# Results

## Transient association of Swi6 with constitutive heterochromatin

*Schizosaccharomyces pombe* encodes two HP1 proteins, Chp2 and Swi6. We have previously shown that RNA binds to Swi6, involving the hinge, CD, and the N-terminus, which impedes binding of full-length Swi6 to an immobilized peptide corresponding to residues 1–20 of a K9 trimethylated histone H3 tail (Keller *et al*, 2012). To study the impact of Swi6-RNA interactions on Swi6 dynamics *in vivo*, we created an *S. pombe* strain in which the endogenous *swi6+* gene was tagged C-terminally with a codon-optimized EGFP tag (Sheff & Thorn, 2004) by homologous recombination. Importantly, the Swi6 and EGFP moieties were separated by a GDGAGLIN linker sequence, rendering the Swi6-EGFP fusion protein fully functional (Keller *et al*, 2013). Furthermore, Swi6-EGFP and endogenous Swi6 were expressed at similar levels within the cell (Fig EV1A and B). Thus, in contrast to previous kinetic measurements performed with N-terminally tagged and ectopically expressed, plasmid-borne GFP-Swi6 (Cheutin *et al*, 2004), our approach allowed us to assess heterochromatin dynamics under near physiological Swi6 expression conditions.

In fluorescence recovery after photobleaching (FRAP) experiments, a fluorescent region is irreversibly bleached by a short laser pulse and the recovery of the fluorescence signal measured over time (Carisey *et al*, 2011). We first performed FRAP on Swi6-EGFP heterochromatic loci in exponentially growing cells and acquired single focal plane images every 60 milliseconds (ms) (Fig 1A and Movie EV1). The bleached regions comprised centromeres, telomeres, or the mating type locus. For proteins bound tightly to chromatin, recovery kinetics are expected to be slow or not detectable, as observed for the telomere-binding protein Taz1 (Keller *et al*, 2012). The fluorescence signal for Swi6-EGFP, however, recovered within 3 s with an average half-recovery time ($t_{1/2}$) of 535 ms (Fig 1B and C). The recovery curve holding the average intensities of 31 FRAP experiments fits best to a one-component model $(y(x) = a(1 - \exp(-bx)))$ with an $R$-squared value of 0.9964 (Fig 1B). Addition of a second component did not significantly improve the fit (Fig EV1D and E). Importantly, FRAP performed with cells that express Swi6-EGFP from a plasmid using the same conditions as Cheutin *et al* (0.4 μg/ml thiamine) resulted in a good fit to a two-component model with $t_{1/2}$ values of the individual components in a physiological range (Fig EV1F–H). These results imply the presence of one major population of Swi6 that binds heterochromatin highly dynamically if expressed from the endogenous locus.

To further test whether Swi6 exists in a single population or as two high- and low-mobility fractions ($F_M$ and $F_I$, respectively), we

**Figure 1. Swi6 is highly mobile.**

A   Schematic of single-plane acquisition FRAP on endogenously tagged Swi6-EGFP. Representative still images of a FRAP experiment are shown. See also Movie EV1. Green sphere: yeast nucleus containing Swi6-EGFP; green dots: heterochromatic Swi6-EGFP; red circle: bleached area.

B   Normalized average intensities (black dots) of FRAP on Swi6-EGFP heterochromatic loci ($n = 31$) fitted to a one-component model (blue line). The dashed lines indicate the final relative intensity that is set to 1 and the fluorescence half-recovery time ($t_{1/2}$).

C   Fluorescence $t_{1/2}$ values of FRAP on Swi6-EGFP heterochromatic loci. The box bounds the interquartile range (IQR) divided by the median, and whiskers extend to a maximum of $1.5 \times$ IQR beyond the box.

D   Schematic indicating the volumes measured to calculate the normalized fluorescence recovery of a locus relative to the entire nucleus.

E   Normalized fluorescence recovery values of FRAP on Swi6-EGFP heterochromatic loci. The box bounds the interquartile range (IQR) divided by the median, and whiskers extend to a maximum of $1.5 \times$ IQR beyond the box.

F   FLIP experiment showing a bleached (FRAP, red circle) and unbleached, adjacent locus (FLIP, blue circle) in the same nucleus. The graph displays average relative fluorescence of FLIP (blue), FRAP (red), and control loci (no bleach, green) in other non-bleached cells.

G   Schematic of a line-scan FRAP experiment. Fluorescence intensities are measured along a line of a non-bleached nucleus (Ctrl nucleus) and a bleached nucleus (FRAP nucleus) in Swi6-EGFP *clr4Δ* cells. Red box: bleached region. Right: raw data example.

H   Average relative intensities of a line-scan FRAP experiment on Swi6-EGFP in heterochromatin (blue line), euchromatin (green line), and *clr4Δ* (red line). The curves have a gliding average of 40.

investigated percentage recovery after photobleaching. $F_M$ is the proportion of bleached proteins replaced by unbleached proteins during the experiment. Thus, $F_M$ at a given locus can be determined by dividing fluorescence intensity at the end of the time lapse ($F_\infty$) by fluorescence intensity before the bleaching event ($F_{initial}$). If a second, static or low-mobility population of Swi6 existed, $F_M$ should be less than 1. However, several experimental artifacts can bias estimation of this ratio and need to be taken into account for a precise determination of $F_M$. First, and consistent with a previous report, we observed that heterochromatin domains are highly mobile in *S. pombe* (Cheutin *et al*, 2004) and give a fainter signal

when moving out of the focal plane. This leads to an underestimation of the total recovery if acquisition is made only at single focal planes (Fig EV1I–L). Second, because 20–35% of total fluorescence in the nucleus is lost when bleaching a single locus, the remaining fluorescence available for recovery must be evaluated. We therefore acquired stacks of the whole nucleus before photobleaching and after recovery and calculated the loss of total fluorescence in the whole nucleus to correct the $F_M$ ratio (Fig 1D). This analysis revealed full recovery of the Swi6-EGFP signal by 3–4 s after photobleaching (Fig 1E), suggesting that there is no static or slow recovering population of Swi6 within heterochromatin.

In summary, Swi6-EGFP expressed from its endogenous locus manifests a single, highly dynamic population on heterochromatic domains with an average $t_{1/2}$ of 535 ms in exponentially growing cells. Although we cannot rule out the possibility that a small population of stably associated Swi6-EGFP molecules remained undetected in our analyses, our results demonstrate that the vast majority of Swi6 rapidly exchanges on heterochromatin.

## Swi6 exchanges between independent heterochromatic domains

Swi6-EGFP is highly concentrated in two to six bright foci, with most of the cells containing three foci (Cheutin *et al*, 2004). The amount of Swi6-EGFP outside these foci is marginal. Hence, full recovery of fluorescence after photobleaching in these foci can only be explained if Swi6-EGFP exchanges between the different heterochromatin domains. Our experimental FRAP setup allowed us also to monitor fluorescence loss in photobleaching (FLIP) in the non-bleached heterochromatic foci. Photobleaching of one heterochromatic focus caused a significant loss of fluorescence in the non-bleached foci (Fig 1F), demonstrating that Swi6 does indeed rapidly exchange not only on heterochromatin *in cis* but also between heterochromatic domains *in trans*.

These results imply that Swi6 very rapidly transfers between heterochromatic domains via the nucleoplasm. Because our standard FRAP setup is limited to image acquisition rates of 1 per 60 ms, we switched to a high-speed confocal line-scan microscopy approach, allowing measurement of fluorescence intensities along a single line with a time resolution of ±2 ms (Fig 1G). The results revealed full fluorescence recovery in < 1 s, with an average $t_{1/2}$ of approximately 89 ms in non-heterochromatic areas of the nucleus (Fig 1H). Even faster Swi6-EGFP kinetics of approximately 49 ms were obtained by repeating the same experiments in cells lacking Clr4, the sole histone methyltransferase that methylates H3K9 in *S. pombe*. As expected, line-scan FRAP in heterochromatic loci revealed slower Swi6-EGFP kinetics than in the nucleoplasm or in *clr4Δ* cells (Fig 1H).

In summary, Swi6 is highly mobile throughout the nucleus, exchanging between different heterochromatic domains via the nucleoplasm. As expected for a heterochromatin-binding protein, Swi6 dynamics are influenced by H3K9 methylation.

## Swi6 *in vivo* dynamics are slowed down by heterochromatic RNA

We previously demonstrated that the kinetic off-rate constant for dissociation of the Swi6-H3K9me3 complex *in vitro* is in the range of 10–1,000/s, corresponding to a lifetime of 1–100 ms (Keller *et al*, 2012). These data are consistent with the $t_{1/2}$ values obtained in our FRAP experiments. Importantly, Swi6-RNA and Swi6-H3K9me3 interactions are mutually exclusive *in vitro* and we therefore predicted that RNA influences Swi6 dynamics. To test this hypothesis, we performed FRAP experiments with cells expressing either wild-type Swi6 or Swi6 that does not bind RNA (NLS-Swi6-EGFP or NLS-Swi6-KR25A-EGFP, respectively). The KR25A mutation in the hinge region of Swi6 does not affect H3K9 binding but abolishes RNA binding and nuclear import of Swi6. Therefore, fusion of an SV40 nuclear localization signal (NLS) to the N-terminus of the Swi6-KR25A is required (Keller *et al*, 2012). The NLS fused to the N-terminus of Swi6 caused an increase in $t_{1/2}$ values from ±535 ms

to ±790 ms (Fig EV2B). In contrast, $t_{1/2}$ values decreased from ±790 ms for NLS-Swi6-EGFP to ±270 ms for NLS-Swi6-KR25A-EGFP (Figs 2A and B, and EV2A and B).

The observed faster recovery kinetics of the Swi6-KR25A mutant strongly suggested that RNA binding to Swi6 slows its mobility. To rule out the possibility that the effect was due to the KR25A mutations *per se* rather than the inability to bind RNA, we monitored fluorescence recovery kinetics of wild-type Swi6 in *cid14Δ* cells. Cid14 facilitates the removal and subsequent degradation of Swi6-bound RNAs (Buhler *et al*, 2007; Keller *et al*, 2012). Consistent with elevated heterochromatic RNA levels upon deletion of *cid14+*, $t_{1/2}$ values for Swi6-EGFP increased significantly in *cid14Δ* relative to *cid14+* cells (Fig 2C and D). However, $t_{1/2}$ values for the NLS-Swi6-KR25A RNA-binding mutant did not differ between *cid14+* and *cid14Δ* cells (Fig 2E and F). These results demonstrate that Swi6-RNA interactions slow Swi6 mobility in the nucleus and suggest that Cid14-mediated release of heterochromatic RNAs from Swi6 is the rate-limiting step in the Swi6 exchange cycle.

## Swi6 dynamics are influenced by RNA throughout the cell cycle at centromeres and are similar at telomeres

The FRAP experiments described could not address the question of whether heterochromatin at centromeres or telomeres could differently influence Swi6 dynamics. Furthermore, Swi6 dynamics might differ in particular phases of the cell cycle. To specifically visualize and photobleach telomeric or centromeric heterochromatin, we generated strains expressing Taz1-mCherry or Cnp1-mCherry, respectively, in addition to Swi6-EGFP (Fig 3A). FRAP experiments on Cnp1-marked centromeric heterochromatin revealed average $t_{1/2}$ values for Swi6-EGFP of approximately 520 ms, very similar to the values obtained from FRAP experiments that were not centromere specific (Figs 1B and 3D). Very similar fluorescence recovery kinetics for telomeric heterochromatin were recorded (Fig 3D) and, thus, both constitutive heterochromatin regions are equally accessible to incoming fluorescent Swi6 protein. Both regions also appear permissive to low-level transcription. Consistent with RNA production, we observed significantly lower $t_{1/2}$ values for NLS-Swi6-KR25A-EGFP than for NLS-Swi6-EGFP at both centromeres and telomeres (Fig 3B and C).

In asynchronous *S. pombe* cultures, most cells are in the G2 phase of the cell cycle. Biochemical experiments performed with unsynchronized populations of *S. pombe* thus mainly represent G2. In contrast, our live cell imaging approach allowed us to look at individual cells representing different cell cycle stages. To examine whether Swi6 dynamics change during the cell cycle, we performed FRAP on individual cells residing in G1, S, or G2. Values of $t_{1/2}$ for NLS-Swi6-EGFP ranged from 300- to 2,200 ms throughout the entire cell cycle (Fig 3E). This high variability of $t_{1/2}$ remained constant over the entire time course. The $t_{1/2}$ values for the RNA-binding mutant NLS-Swi6-KR25A also remained the same throughout the entire cell cycle. However, the dynamic range of the $t_{1/2}$ values for NLS-Swi6-KR25A-EGFP was much smaller than for NLS-Swi6-EGFP (Fig 3E). Interestingly, the average $t_{1/2}$ values for NLS-Swi6-EGFP in G2 and S cells were not significantly different but in G1 were significantly lower (Fig 3F). Moreover, Swi6 mobility was greater in the absence of RNA binding, including the cells residing in G2 (Fig 3E and G).

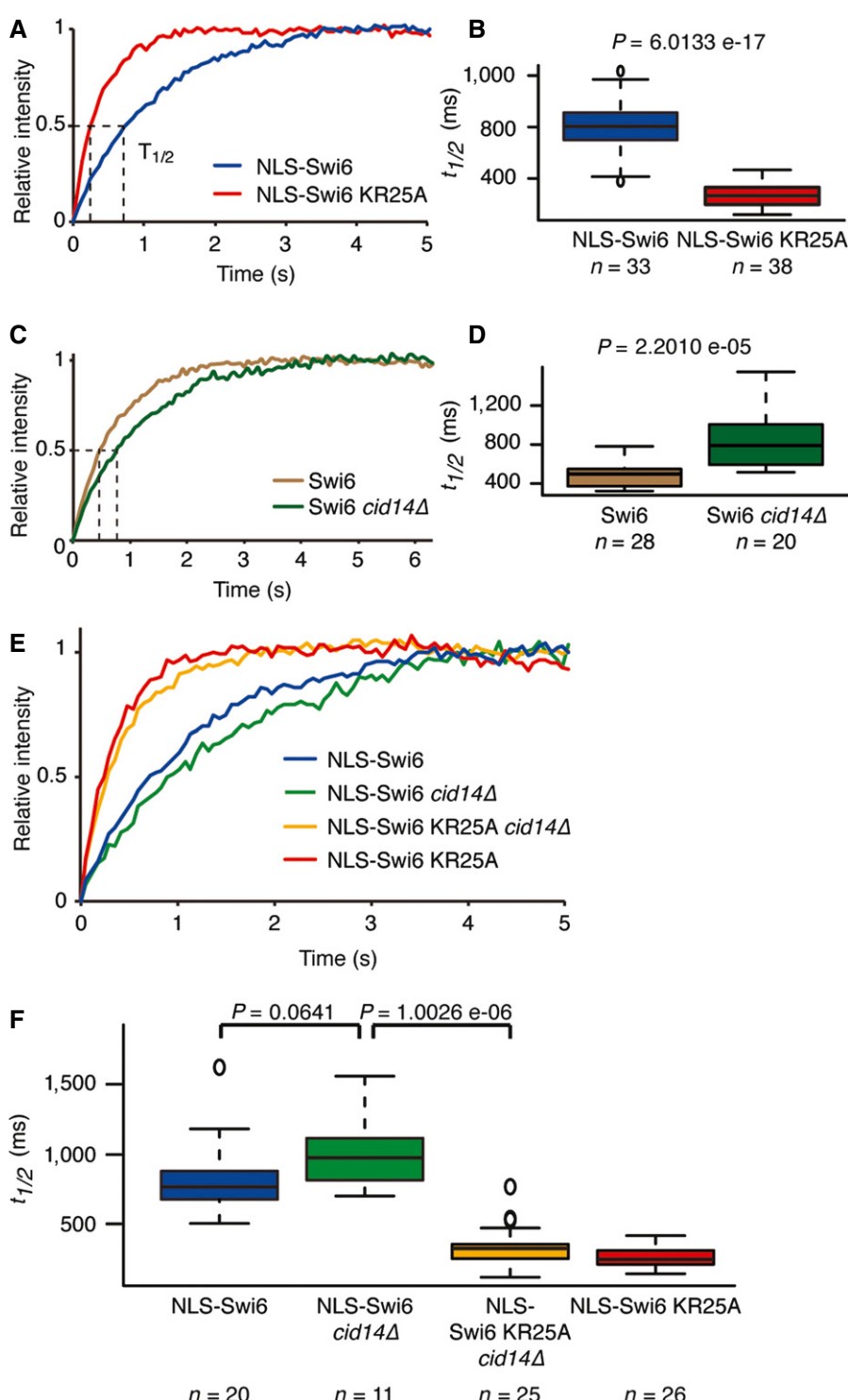

**Figure 2. RNA decelerates Swi6 dynamics.**

A, B    Average relative intensities over time and corresponding fluorescence $t_{1/2}$ values of heterochromatic Swi6 obtained from FRAP experiments performed with cells expressing NLS-Swi6-EGFP (blue) or NLS-Swi6-KR25A-EGFP (red).

C, D    Average relative intensities over time and corresponding fluorescence $t_{1/2}$ values of heterochromatic Swi6 obtained from FRAP experiments performed with wild-type (brown) or *cid14Δ* (dark green) cells expressing Swi6-EGFP.

E, F    Average relative intensities over time and corresponding fluorescence $t_{1/2}$ values of heterochromatic Swi6 obtained from FRAP experiments performed with wild-type cells expressing heterochromatic NLS-Swi6-EGFP (blue) or NLS-Swi6 KR25A-EGFP (red), or *cid14Δ* cells expressing NLS-Swi6-EGFP (green) or NLS-Swi6-KR25A-EGFP (yellow).

Data information: In (B), (D) and (F), the box bounds the counterquartile range (IQR) divided by the median, and whiskers extend to a maximum of 1.5 × IQR beyond the box.

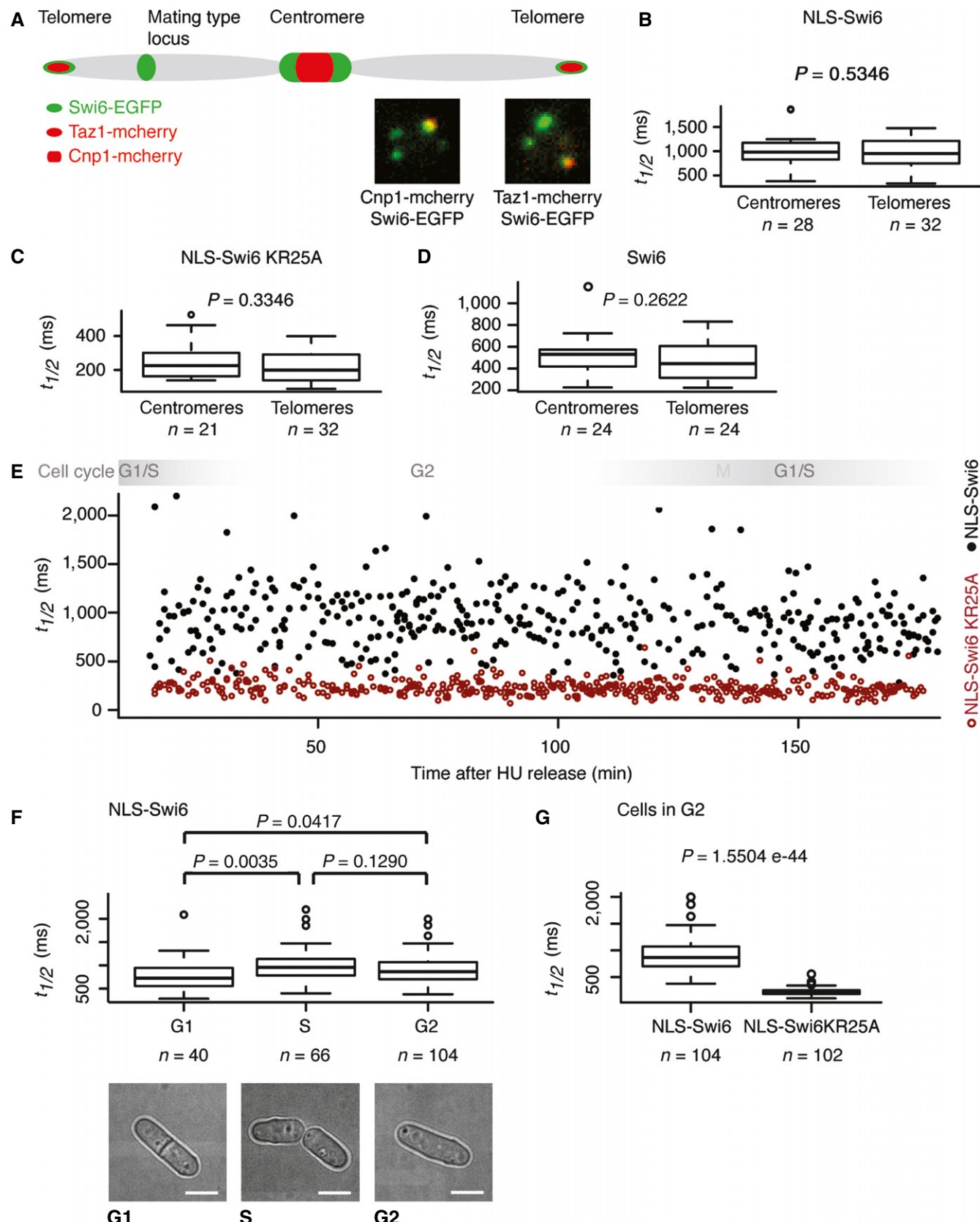

**Figure 3.**

**Figure 3.  Swi6 dynamics remain the same throughout the cell cycle.**

A   Schematic of a chromosome indicating Swi6-EGFP heterochromatic domains (green). Cnp1-mCherry and Taz1-mCherry (red) mark centromeres and telomeres, respectively. Representative images demonstrating Swi6-EGFP co-localization with centromeres (Cnp1-mCherry) and telomeres (Taz1-mCherry) are shown.

B   Fluorescence $t_{1/2}$ values of centromeric or telomeric Swi6 obtained from FRAP experiments performed with cells expressing NLS-Swi6-EGFP.

C   Fluorescence $t_{1/2}$ values of centromeric or telomeric Swi6 obtained from FRAP experiments performed with cells expressing the RNA-binding mutant NLS-Swi6-KR25A-EGFP.

D   Fluorescence $t_{1/2}$ values of centromeric or telomeric Swi6 obtained from FRAP experiments performed with cells expressing Swi6-EGFP.

E   Fluorescence $t_{1/2}$ values of centromeric Swi6 obtained from FRAP experiments performed after release of cells expressing NLS-Swi6-EGFP (black dots) or the RNA-binding mutant NLS-Swi6-KR25A-EGFP (red dots) from G1/S cell cycle arrest. Each dot represents one FRAP experiment at the respective time after release from cell cycle arrest. Swi6 is dispersed in M phase due to H3S10 phosphorylation, and M phase was therefore not included in the FRAP analysis.

F   Fluorescence $t_{1/2}$ values obtained from FRAP experiments performed on centromeric Swi6. A total of 40, 66, and 104 individual cells residing in the G1, S, and G2 phases of the cell cycle were analyzed, respectively. Representative bright-field images of cells in G1, S, or G2 are shown. Scale bars = 2 μm.

G   Fluorescence $t_{1/2}$ values obtained from FRAP experiments performed on centromeric Swi6 of cells expressing NLS-Swi6-EGFP and NLS-Swi6-KR25A-EGFP that reside in the G2 of the cell cycle.

Data information: In (B–D), (F) and (G), the box bounds the interquartile range (IQR) divided by the median, and whiskers extend to a maximum of 1.5 × IQR beyond the box.

Together these results show that Swi6 dynamics in heterochromatin do not differ significantly at centromeres compared with telomeres and respond equally to RNA binding at these heterochromatin regions. Thus, similar to centromeric repeats, telomeric heterochromatin also appears permissive for RNA synthesis. Consistent with this, transcripts originating from the subtelomeric *tlh1+* gene were clearly detected in wild-type cells (Fig EV1C). Interestingly, Swi6 mobility is markedly influenced by RNA binding throughout the cell cycle, suggesting that heterochromatic RNA production is not restricted to G1/S. Yet, the high dynamic range and random distribution of $t_{1/2}$ values for wild-type Swi6 indicate widely fluctuating and stochastic production of RNA within *S. pombe* heterochromatin.

**Swi6 dimerization prevents spreading of centromeric H3K9 methylation**

A characteristic, conserved feature of heterochromatin is that it spreads along chromatin from specific nucleation sites (Grewal & Moazed, 2003). Current models for heterochromatin spreading in *S. pombe* assign Swi6 a central role. It is generally assumed that it acts via a stepwise higher-order oligomerization process and recruitment of histone-modifying activities (Canzio *et al*, 2014). Therefore, our observation that Swi6 exists as one single dynamic population in heterochromatin is surprising, as a slowly exchanging fraction of Swi6 would be expected upon formation of an oligomeric network. We reasoned that such stably bound Swi6 molecules are at very low abundance and remained undetected in our FRAP experiments.

If a minor subpopulation of Swi6 that promotes spreading by oligomerization existed, defects in the spreading of H3K9 methylation in the absence of Swi6 would be expected. To test this hypothesis, we performed ChIP sequencing (ChIP-seq) with *swi6+*

and *swi6Δ* cells and an antibody specifically recognizing H3K9me2. Consistent with the above hypothesis, we observed partially disrupted H3K9me2 in subtelomeric regions (Fig EV3A), as well as partial loss of H3K9me2 at the silent mating type locus, as previously reported (Hall *et al*, 2002) (Fig EV3B). However, in contrast to *chp2Δ* cells (Fig EV4B), levels of H3K9me2 were not reduced at any of the pericentromeric repeat regions in cells lacking Swi6 (Figs 4A and EV3E and F). This is consistent with a previous report of no significant reduction in H3K9me2 levels at specific centromeric sites in *swi6Δ* cells as measured by regular ChIP–PCR (Sadaie *et al*, 2004). However, we observed unexpected spreading of H3K9me2 beyond normal boundaries of heterochromatin in the absence of Swi6, most pronounced at the inverted repeat element IRC1R boundary region of centromere 1 (Fig 4A) and in the subtelomeric region of the right arm of chromosome 1 (Fig 4B). Similarly, we observed increased H3K9 methylation levels on the subtelomeric *tlh1+* and *tlh2+* genes in *swi6Δ* cells (Fig EV3A and D).

Thus, Swi6 is not obligatory for spreading of centromeric heterochromatin but is required for the demarcation of heterochromatin at the IRC1 boundaries of centromere 1. This is highly unexpected and raises the question of whether we are seeing an indirect consequence of the complete loss of Swi6 or whether Swi6 acts *per se* in blocking the propagation of H3K9 methylation. Therefore, we assessed by ChIP–PCR H3K9 methylation at the IRC1R boundary in cells that express Swi6 bearing single point mutations in either the CSD (*swi6L315E*), which abolish dimerization of Swi6 (Cowieson *et al*, 2000; Canzio *et al*, 2011), or the CD (*swi6W104A*), which abolish H3K9me2 binding (Jacobs & Khorasanizadeh, 2002; Canzio *et al*, 2013). Consistent with our ChIP-seq results, H3K9me2 in the centromeric repeat region (dh/dg) was not significantly different in wild-type and *swi6Δ* cells but was markedly increased on the neighboring *emc5+* and *rad50+* genes (Fig 5B and C). Likewise, H3K9me2

**Figure 4.  Methylation of H3K9 spreads across natural boundaries and heterochromatin becomes derepressed in *swi6Δ* cells.**

A   H3K9me2 ChIP-seq enrichment profiles for *swi6+* and *swi6Δ* cells on centromere 1. Centromeric repeat elements (dg/dh) are indicated. Inverted repeats (IRC1) constitute heterochromatin boundaries in wild-type cells (Cam *et al*, 2005). The *y*-axes represent log₂ ChIP-seq enrichments in 200-bp genomically tiled windows calculated over *clr4Δ* cells. Two independent biological replicates for each genotype were performed (replicates 1 and 2). Positions of genomic elements on the plus and minus strands are indicated. Blue, protein-coding genes; white, repeats; green, long terminal repeat (LTR).

B   H3K9me2 ChIP-seq enrichment profiles for *swi6+* and *swi6Δ* cells on the telomere of the right arm of chromosome 1. Two independent biological replicates for each genotype were performed (replicates 1 and 2).

C   Differential expression of transcripts coming from centromere 1 assessed by tiling microarray gene expression analysis. Expression data for *swi6+* and *swi6Δ* cells were taken from a previous publication (Woolcock *et al*, 2012).

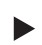

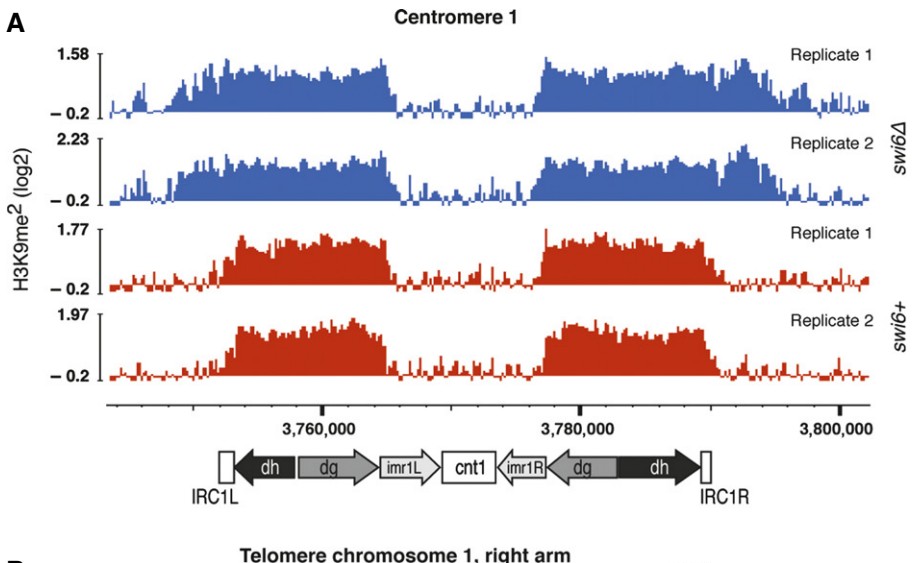

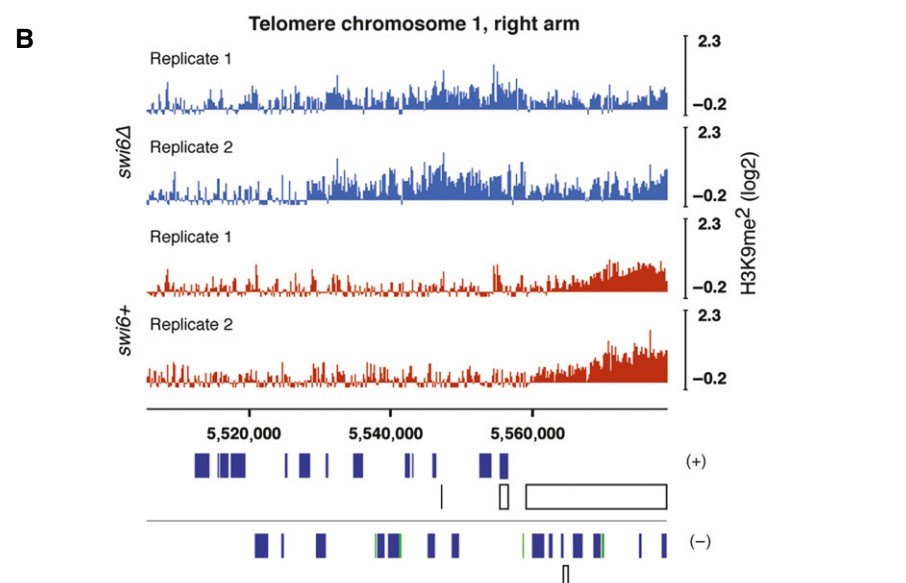

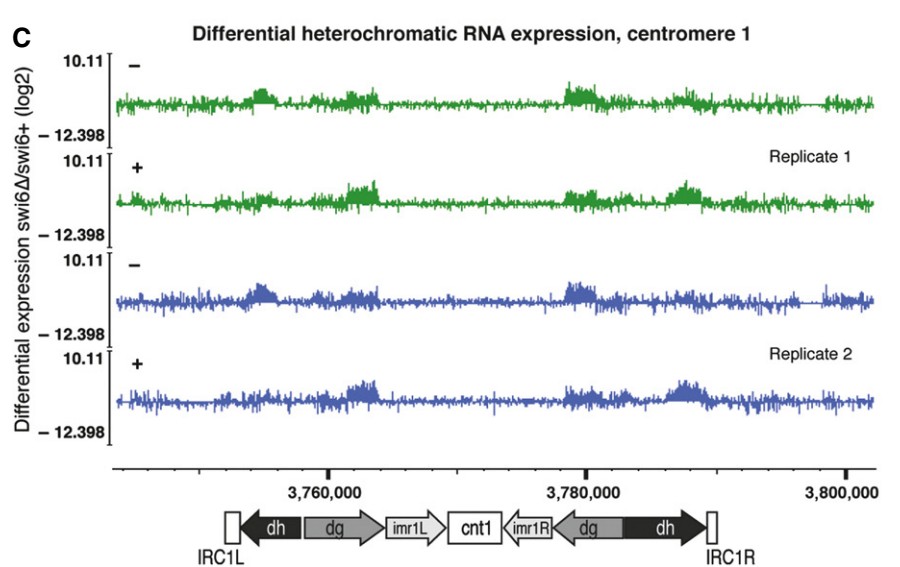

**Figure 4.**

spread beyond the IRC1R boundary in both *swi6L315E* and *swi6W104A* cells (Fig 5A–D). Hence, Swi6 dimerization and H3K9me2 binding are both critical for limiting the spreading of H3K9 methylation.

The results so far indicate that Swi6 is directly involved in counteracting spreading of H3K9 methylation. Importantly, the anti-silencing factor Epe1 has been reported also to limit the spreading of H3K9 methylation (Ayoub *et al*, 2003; Zofall & Grewal, 2006; Trewick *et al*, 2007). Thus, as Epe1 recruitment to heterochromatin depends on Swi6, spreading of H3K9 methylation in the absence of Swi6, or Swi6 binding to H3K9me2, could be explained by failure to recruit Epe1 (Zofall & Grewal, 2006). To test this hypothesis directly, we measured H3K9me2 levels in wild-type, *epe1Δ,* and *swi6Δ* cells. The levels of H3K9me2 in *epe1Δ* were slightly higher than in wild-type cells but not statistically significant and never as high as H3K9me2 in *swi6Δ* cells (Fig 5E). Therefore, spreading of H3K9 methylation in the absence of Swi6 cannot be explained solely by impaired Epe1 recruitment.

In conclusion, our results reveal that Swi6 dimerization is crucial for the prevention of H3K9 methylation spreading beyond the IRC1R boundary region of centromere 1. Although we cannot rule out Swi6-mediated recruitment of further anti-silencing factors that may act redundantly with Epe1, we favor a model in which heterochromatin spreading is counteracted by a previously described distinct conformational state of Swi6 dimers (see Discussion).

## Tas3 self-association is required for H3K9 methylation in the absence of Swi6

Besides that of the precise contribution of Swi6 to the restriction of H3K9me2 to its natural boundaries, the question arises as to what mechanism propagates H3K9me2 beyond the IRC1R boundary in the absence of Swi6. To test the possible involvement of the RNAi pathway, which is essential for the assembly and maintenance of centromeric heterochromatin, we profiled small interfering RNAs (siRNAs) in wild-type and *swi6Δ* cells by deep sequencing. Instead of finding additional siRNAs mapping to the heterochromatin flanking regions, we observed slightly reduced levels of canonical centromeric siRNAs and loss of border RNAs (brdrRNAs) (Keller *et al*, 2013) that map to the IRC1R boundary (Fig 6A).

The absence of siRNAs mapping to the region that becomes H3K9 methylated in *swi6Δ* cells argues against the deposition of

H3K9me2 through the canonical RNAi-mediated heterochromatin assembly pathway (Holoch & Moazed, 2015). However, the RNA-induced transcriptional silencing (RITS) complex was shown previously to spread from siRNA-producing centromeric heterochromatin nucleation sites to regions with few or no siRNAs (Li *et al*, 2009). Therefore, we hypothesized that spreading of H3K9me2 in the absence of Swi6 is caused by spreading of the RITS complex independently of siRNAs. RITS spreading *in cis* depends on

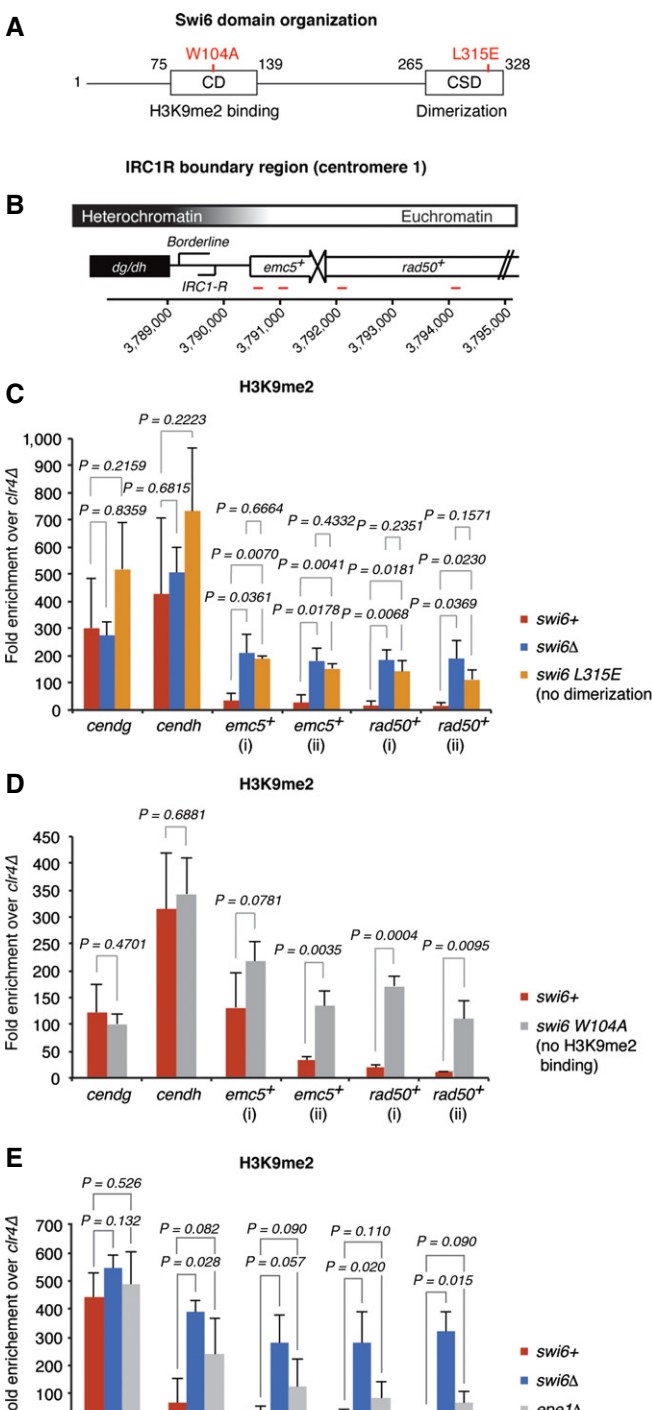

**Figure 5.  Swi6 dimerization and H3K9me2 binding are required for the prevention of H3K9me2 spreading.**

A     Domain architecture of full-length Swi6. The indicated mutations in red disrupt the dimerization property of the CSD (L315E) or the ability of the CD to bind H3K9me2 (W104A).

B     Schematic of the IRC1R boundary region. IRC1R partially overlaps with the region that expresses the noncoding RNA borderline (Keller *et al*, 2013). Positions of the quantitative RT–PCR primers are indicated in red.

C–E  H3K9me2 levels on heterochromatin-adjacent genes assessed by ChIP–PCR in *swi6+*, *swi6Δ*, and *swi6-L315E* (CSD dimerization mutant) (C), *swi6-W104A* (H3K9me2 binding mutant) (D), and *epe1Δ* (E) cells. Enrichments over *clr4Δ* are normalized to *adh1+*. Average fold enrichment with s.d. is shown for three (C and E) or four (D) independent experiments. *P*-values were generated by the Student's *t*-test (two-tailed distribution, two-sample, unequal variance).

a C-terminal Tas3 α-helical motif (TAM) that can undergo polymeric self-association *in vitro* (Li *et al*, 2009). Because Tas3–Tas3 interactions mediated by TAM have been proposed to be especially important for RITS spreading into regions with few or no siRNAs, we speculated that Tas3-TAM plays a critical role in H3K9me2 spreading from centromeric repeats into the region flanking the IRC1R boundary in the absence of Swi6. To test this, we introduced a single amino acid substitution (L479E, Fig 6B) that abrogates Tas3–TAM polymerization (Li *et al*, 2009) in *swi6+* and *swi6Δ* cells and assessed H3K9me2 levels by ChIP (Fig 6C). H3K9me2 levels were mildly affected in *tas3-L479E* cells but completely eradicated in *tas3-L479E swi6Δ* double-mutant cells (Fig 6D). Thus, upon removal of Swi6, Tas3 self-association becomes absolutely essential to the maintenance of H3K9 methylation within centromeric repeat sequences. This strongly suggests that centromeric H3K9me2 spreading in *swi6Δ* cells is mediated by self-association of Tas3. The importance of Tas3 self-association to the maintenance of H3K9 methylation at other heterochromatin regions awaits further investigations.

## Discussion

In this study, we have reanalyzed the dynamics of the *S. pombe* HP1 protein Swi6 in living cells and investigated its contribution to the maintenance and spreading of heterochromatin. We found that Swi6 rapidly exchanges *in cis* and *in trans* between different heterochromatic regions and that Swi6 mobility is decelerated by methylated histone H3K9 and RNA. Although we show that Swi6 could in principle exist in at least three kinetically distinct populations (Fig 7), we found that most if not all Swi6 molecules exchange between the RNA and heterochromatin-bound form under physiological conditions. Our results are consistent with a stochastic model of heterochromatin (Cheutin *et al*, 2004) and suggest that (i) heterochromatin in *S. pombe* is permissive for transcription throughout the cell cycle, (ii) only very few Swi6 molecules are engaged in heterochromatin spreading by oligomerization at telomeres, and (iii) Swi6 is dispensable to the expansion of centromeric heterochromatin. Rather, it appears to be involved in the demarcation of centromeric heterochromatin from neighboring euchromatin by a mechanism that remains to be fully elucidated. We discuss below the implications of these findings for the mechanisms of heterochromatin formation and repression in fission yeast.

### The role of Swi6 in maintenance and spreading of heterochromatin

Iterative HP1 binding to methylated H3K9 and recruitment of HMTase activity is the prevalent model for heterochromatin spreading (Eissenberg & Elgin, 2014). The involvement of Swi6 in heterochromatin spreading is strongly supported by *in vivo* experiments using Swi6 overexpression to force the expansion of heterochromatin (Noma *et al*, 2006; Wang *et al*, 2013). *In vitro* data on Swi6-nucleosome interactions further imply a direct role of Swi6 in spreading along the chromatin fiber through a process of stepwise higher-order oligomerization (Canzio *et al*, 2011, 2013). The Swi6 dynamics determined in this study imply that such

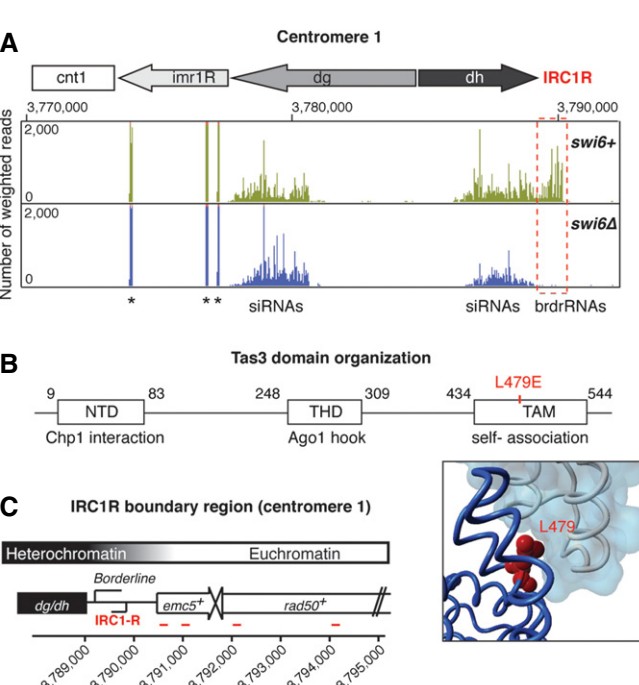

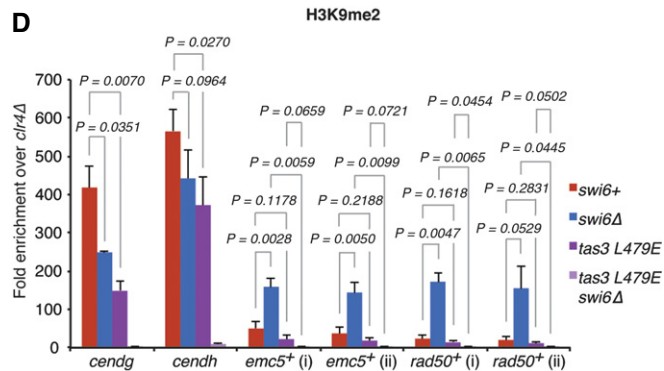

**Figure 6. Tas3 self-association is required for H3K9 methylation at centromeric repeats in the absence of Swi6.**

A  sRNA reads mapping to centromere 1 in *swi6+* (green) and *swi6Δ* (blue) cells. The dashed red box highlights the loss of brdrRNAs at the IRC1R in the absence of Swi6. Centromeric repeat elements and the central core are indicated. Counts were normalized to the library size. Asterisks denote tRNA fragments. Note that tRNA genes flank IRC3 (Fig EV4) but not IRC1R.

B  Domain architecture of full-length Tas3, adapted from Li *et al* (2009). The N-terminal domain (NTD) and the Tas3-homology domain (THD) are responsible for the interactions with Chp1 and Ago1, respectively. Tas3 self-association is mediated through the TAM motif, which is disrupted by the point mutation L497E, indicated in red. The location of residue L479 (red) is highlighted in the tube representation of two Tas3 molecules (dark blue and gray) with one surface view (light blue). Created from structure 3D1D of the PDB database.

C  Schematic of the IRC1R boundary region. IRC1R partially overlaps with the region that expresses the noncoding RNA borderline (Keller *et al*, 2013). The positions of the primers used for quantitative RT–PCR are indicated in red.

D  H3K9me2 levels on heterochromatin-adjacent genes assessed by ChIP–PCR in *swi6+*, *swi6Δ*, *tas3 L479E*, and *tas3 L479E swi6Δ* cells. Enrichments over *clr4Δ* are normalized to *adh1+*. Average fold enrichment with s.d. is shown for three independent experiments. *P*-values were generated by Student's *t*-test (two-tailed distribution, two-sample, unequal variance).

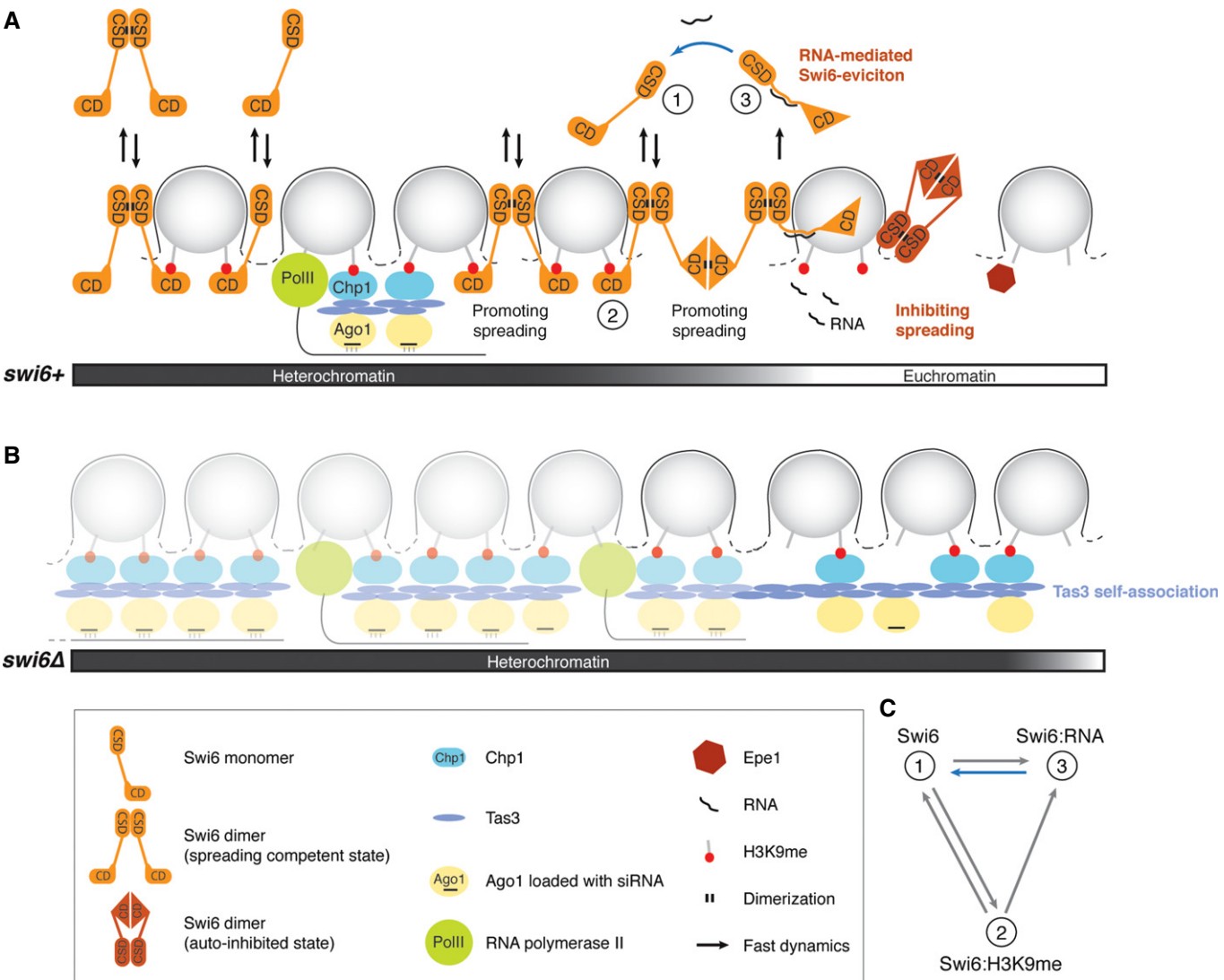

**Figure 7.  Regulation of Swi6 dynamics and the role of Swi6 in heterochromatin spreading, boundary formation, and repression.**

A   Model for the formation of a distinct heterochromatic domain. The spreading-competent open conformation of Swi6 dimers is fully dispensable or acts redundantly with RNAi-mediated spreading mechanisms in the formation of constitutive heterochromatin. The spreading of heterochromatin into neighboring euchromatin in the absence of DNA-encoded boundary elements, such as TFIIIC binding sites, can be stopped by the spreading-incompetent closed conformation of Swi6 dimers, RNA-mediated eviction of spreading-competent Swi6, or Epe1 activity. Swi6 molecules interact with H3K9me2 only transiently and exchange between free and RNA-bound forms (2, 1, or 3, respectively; see also C). Removal of RNA from Swi6 constitutes a rate-limiting step in the Swi6 exchange cycle (blue arrow) and contributes to tight repression of heterochromatin.

B   In the absence of Swi6, H3K9 methylation is mediated by the RITS complex (Ago1, Chp1, Tas3) and can spread beyond natural boundaries via self-associating Tas3. This might occur independently of siRNAs.

C   In theory, Swi6 can exist in at least three kinetically distinct populations: 1) freely diffusible Swi6, 2) Swi6 bound to H3K9me2, and 3) Swi6 bound to RNA; 3 is the least and 1 the most mobile population. The blue arrow indicates the rate-limiting step in the Swi6 exchange cycle. Under physiological conditions, most Swi6 molecules exchange between the RNA- and H3K9me2-bound forms.

oligomeric networks, if indeed they exist *in vivo* under natural conditions, are not inert but rather exist as a steady-state equilibrium of association and dissociation of Swi6 molecules with nucleosomes and themselves. We postulate that only very few if any Swi6 molecules remain stably bound to chromatin under physiological conditions; the majority are in a dynamic equilibrium. Only Swi6 overexpression results in stable association with chromatin and may enhance heterochromatin spreading. This may

explain why strains with additional copies of the *swi6+* gene were employed in previous investigations of heterochromatin boundaries (Noma *et al*, 2001, 2006; Wang *et al*, 2013; Verrier *et al*, 2015).

Our ChIP-seq results suggest that Swi6-mediated spreading contributes partially to the propagation and spreading of the H3K9me2 mark at some telomeric regions of *S. pombe* chromosomes under physiological conditions. In contrast, Swi6 is fully

dispensable to the maintenance of H3K9me2 at centromeres. However, we found that Swi6 is required for the restriction of H3K9me2 to the pericentromeric repeat region and the prevention of spreading into neighboring euchromatin. This unanticipated finding highlights the functional promiscuity of HP1 proteins and raises the question of how Swi6 participates in counteracting heterochromatin spreading. Our results show that it cannot simply be recruitment of the anti-silencing factor Epe1, because H3K9 methylation in *epe1Δ* cells does not phenocopy the spreading we observed in *swi6Δ* cells (Fig 5E). This is further supported by a recent study demonstrating that H3K9me2 levels around the IRC1 boundary regions of centromere 1 in *epe1Δ* cells are similar to those of wild-type cells, unless Swi6 is overexpressed (Wang *et al*, 2013). Furthermore, Epe1 binding at the IRC1 boundary element was reported in Swi6-defective cells (Zofall & Grewal, 2006). Thus, the spreading we observed in *swi6Δ* cells cannot be explained by a failure to recruit Epe1 to the boundary. Therefore, further properties of Swi6 are likely to contribute to boundary formation. Intriguingly, Swi6 dimers that form through CSD interactions have recently been demonstrated to exist in closed and open states *in vitro*. The closed state is mediated by interaction between the two CDs of a Swi6 dimer and inhibits binding of the H3K9me2 mark. Furthermore, this conformation cannot participate in the Swi6 oligomerization process and could therefore stop spreading of heterochromatin (Canzio *et al*, 2013). Supporting such a model, we found that dimerization is crucial to Swi6 activity in boundary protection (Fig 5C). It is possible that proteins bound at heterochromatin–euchromatin transitions can induce and stabilize the closed state of Swi6 dimers and thereby stop encroachment of heterochromatin into neighboring regions. Similarly, RNA could also drive Swi6 into the closed state. Consistent with this model is our previous observation that heterochromatin spreads into neighboring euchromatin if Swi6 fails to bind RNA (Keller *et al*, 2013). Thus, in addition to evicting spreading-competent Swi6 from the chromatin template, as we have previously suggested, ncRNAs may also switch Swi6 into the closed Swi6 conformation that stops the spreading of heterochromatin. This is an attractive hypothesis that demands further study.

We propose a model in which RNAi- and Swi6-dependent mechanisms act redundantly within centromeric repeats and maintain heterochromatin under physiological conditions (Fig 7A). Redundancy between the two mechanisms is supported by a previous study, which showed that Swi6 maintains some H3K9 methylation at centromeric repeats in the absence of RNAi (Sadaie *et al*, 2004). Similarly, we observed only slightly reduced H3K9me2 levels in *tas3L479E* and *swi6Δ* cells, but complete loss of H3K9me2 in *tas3L479E swi6Δ* double-mutant cells (Fig 6D). We envision that spreading-incompetent (closed) Swi6 dimers is an important factor in counteracting the spread of heterochromatin into the repeat proximal regions, which can be mediated either by the open conformation of Swi6 dimers or by RNAi-mediated H3K9me2 deposition. In the absence of Swi6, additional H3K9me2-binding sites are available for the Chp1 subunit of the RITS complex, which stabilizes the RNAi machinery on chromatin and thereby promotes the methylation of neighboring nucleosomes (Fig 7B). Importantly, our results highlight the importance of Tas3 self-association for such spreading of H3K9me2 and that this may occur independently of siRNAs (Fig 6).

## The role of Swi6 in silencing of heterochromatin

Whereas Swi6 is dispensable to the maintenance of H3K9me2 at centromeric repeats, it is required for full repression. Previous quantitative RT–PCR analyses have demonstrated that Swi6 contributes roughly 14% and 32% to the silencing of centromeric *dg* and *dh* sequences, respectively. At telomeres of *S. pombe* chromosomes, Swi6 was shown to be required for the majority of silencing at the subtelomeric *tlh1+* gene (Motamedi *et al*, 2008). These findings are in agreement with our high-resolution gene expression data and demonstrate that complete repression of heterochromatin cannot be achieved without Swi6 (Figs 4C and EV5) (Woolcock *et al*, 2012).

We previously postulated a model for the silencing of heterochromatin in which Swi6 serves a general function linking transcription to downstream RNA degradation, rather than reducing transcription *per se*. Importantly, we have demonstrated that Swi6 complexed with RNA dissociates from H3K9-methylated nucleosomes and escorts its associated RNAs to the RNA-decay machinery, thereby contributing to tight repression of heterochromatin at a co- or post-transcriptional level (Keller *et al*, 2012). Our current study extends these findings and reveals that RNA binding to Swi6 decelerates the mobility of Swi6 on heterochromatin about three fold, suggesting that release of heterochromatic RNAs from Swi6 is the rate-limiting step in the Swi6 exchange cycle (Fig 7A and C).

In conclusion, rather than promoting the maintenance and spreading of heterochromatin, ensuring tight repression of heterochromatin seems to be the prevailing activity of Swi6. Our results are consistent with a model for heterochromatin silencing in which Swi6 assures coupling between heterochromatin transcription and RNA degradation by acting as an H3K9me2-specific checkpoint. Molecular details of RNA decay downstream of Swi6 remain elusive, and future investigations focused on RNA decay are thus required in order to achieve a better mechanistic understanding of heterochromatin silencing.

# Materials and Methods

### Strains and plasmids

Fission yeast strains and plasmids used in this study are described in Appendix Tables S1 and S2.

### Western blot and antibodies

Total proteins from exponentially growing cells were extracted using TCA and separated on a NuPAGE 4–12% Bis–Tris gel (Invitrogen). Antibodies for Western blotting were used at the following concentrations: Swi6 (in-house generated; 1:20,000), tubulin ((Woods *et al*, 1989), 1:3,000). The Swi6 mouse monoclonal antibody was raised against full-length recombinant Swi6 and affinity-purified with Protein G.

### Fluorescence recovery after photobleaching (FRAP)

Imaging was performed with an Olympus IX81 microscope equipped with a Yokogawa CSU-X1 spinning disk, a PlanApo 100×/1.45 TIRFM oil objective, two back-illuminated EM-CCD EvolveDelta

cameras (Photometrics, AZ), 491-nm and 561-nm laser lines (Cobolt, Sweden), a Semrock Di01-T488/568 dichroic, and Semrock FF01-525/40-25 and FF01-440/521/607/700-25 emission filters. All devices were piloted with the software Visiview (Visitron GmbH, Puchheim, Germany). For FRAP experiments, a VisiFRAP module (Visitron), a 473-nm laser line and a chroma Z405/473rpc-xt dichroic were installed on the setup. The bleach region was a diffraction-limited spot, and the bleach time was 20 ms.

For each FRAP experiment, a time series of 120 images of a fixed confocal plane was acquired every 60 ms, while the bleach pulse was aimed and triggered manually by mouse click. Additionally, in order to calculate the recovery percentage of the bleached foci, a stack was taken before and after the time series acquisition. All images were acquired at 30°C. Cells were grown in YES medium to exponential phase and imaged on a slide harboring an agarose patch containing YES medium with 3% glucose.

### FRAP during the cell cycle

Exponentially growing cells were synchronized in G1/S phase by a 4-h treatment with 15 mM hydroxyurea and released into the cell cycle after a wash with YES medium. The cells were imaged in a Ludin Chamber with a lectin-coated glass slide (BS-I; Sigma). FRAP experiments were performed for approximately 3 h after release, each time-point representing the recovery curve of a locus from a different cell. All images were acquired at 30°C. M phase was not included in the FRAP analysis as Swi6 becomes dispersed (Ekwall *et al*, 1995; Pidoux *et al*, 2000; Li *et al*, 2013).

### Line-scan FRAP

Imaging was performed on an AXIO OBSERVER Z1 equipped with an LSM 710 scanning head, a multiline Argon 458/488/514 nm (25 mW) laser and a Plan-Apochromat 63×/1.40 Oil DIC M27 objective. The devices were piloted with ZEN Black 2010 software.

### Analysis of FRAP data

The acquired images were analyzed using custom script in the open-source software Fiji (Schindelin *et al*, 2012). For each FRAP time series, we manually assigned the bleached region as a region of interest (ROI) and calculated the mean intensity of the ROI. We subtracted the obtained minimal intensity from these mean intensities. To calculate the recovery half-times ($t_{1/2}$), we performed an exponential one-component curve fit based on the formula $y(x) = a(1 - \exp(-bx))$. The boxplots of the $t_{1/2}$ values were made in RStudio.

### Curve fitting of mean recovery curves

To obtain mean recovery curves of the different conditions, we normalized each individual series of intensities to the mean of its last 15 time-points post-recovery. Of these normalized intensity series, aligned to the bleaching time-point, the means were calculated and loaded into the curve fitting toolbox of Matlab (MATLAB and Curve Fitting Toolbox Release 2013b; The MathWorks, Inc., Natick, USA). The one-component fits were based on the formula $y(x) = a(1 - \exp(-bx))$, while the two-component fits were based on $y(x) = a(1 - \exp(-bx)) + c(1 - \exp(-dx))$.

### Chromatin immunoprecipitation (ChIP)

H3K9me2 ChIP experiments were performed with an H3K9me2-specific mouse monoclonal antibody from Wako (clone no. MABI0307; 302-32369). Anti-H3K9me antibody was used at 1 μg per mg of whole-cell extract (WCE). Cells were processed for ChIP analysis as previously described (Keller *et al*, 2013).

### ChIP sequencing (ChIP-seq)

ChIP-seq libraries were generated with an Illumina-based protocol with custom reagents and bar-coded adapters as previously described (Keller *et al*, 2013). Libraries were sequenced on the Illumina HiSeq2000 system according to the manufacturer's protocols.

### sRNA sequencing

RNA for sRNA sequencing was prepared as previously described (Keller *et al*, 2013). Libraries were prepared with the Illumina TruSeq sRNA preparation protocol (Cat. no. RS-930-1012). The 145- to 160-nt population was isolated and the library sequenced on an Illumina HiSeq2000. sRNA reads were aligned as described previously (Emmerth *et al*, 2010) with zero mismatch allowed.

### Accession codes

The deep sequencing data are deposited under the accession code GSE70946 (NCBI Gene Expression Omnibus), which include the ChIP-seq data of the *swi6Δ* samples under the accession number GSE61136 and the small-RNA sequencing data of *swi6+* and *swi6Δ* samples under GSE70945. The ChIP-seq data of wild type were previously published and can be found under accession code GSE42850.

Expanded View for this article is available online:
http://emboj.embopress.org

## Acknowledgements

We would like to thank Yukiko Shimada for excellent technical assistance, Katarzyna Kowalik for bioinformatics advice, Jeffrey Chao and Susan Gasser for constructive advice, Alex Tuck and Pat King for critical reading and editing of the manuscript, and Daniele Oberti and other laboratory members for fruitful discussions. We are grateful to T. Roloff, S. Dessus-Babus, D. Gaidatzis, and H.-R. Hotz for help with Illumina deep sequencing and advice on data analysis. We thank Danesh Moazed and Hisao Musakata for providing *S. pombe* strains. This work was supported by funds from the Swiss National Science Foundation (PP00P3_139204/1). R.K.-S. was supported by a postdoctoral fellowship from the Peter and Traudl Engelhorn Foundation. The Friedrich Miescher Institute for Biomedical Research is supported by the Novartis Research Foundation.

## Author contributions

RS designed and carried out most of the experiments. RK-S performed ChIP, generated libraries for next-generation sequencing, and analyzed ChIP-seq. CK prepared RNA for sRNA deep sequencing. MAK and LG assisted with data analysis and establishment of the methods. MB obtained funding, designed experiments, and oversaw the study. MB and RS wrote the manuscript.

## Conflict of interest

The authors declare that they have no conflict of interest.

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
