## [Review Process File · The EMBO Journal]

Manuscript EMBO-2015-91320

H3K9 methylation extends across natural boundaries of heterochromatin in the absence of an HP1 protein

Rieka Stunnenberg, Raghavendran Kulasegaran, Claudia Keller, Moritz A. Kirschmann, Laurent Gelman and Marc Buehler

Corresponding author: Marc Buehler, Friedrich Miescher Institute For Biomedical Research

Review timeline:

Submission date:	17 February 2015
Editorial Decision:	09 April 2015
Revision received:	28 July 2015
Editorial Decision:	23 August 2015
Revision received:	28 August 2015
Accepted:	03 September 2015

Transaction Report:

Editor: Anne Nielsen

1st Editorial Decision

09 April 2015

1st editorial decision 9th April 2015

Thank you again for submitting your manuscript for consideration here and my sincere apologies for the highly unusual duration of the review process. Your manuscript has now been seen by three referees whose comments are included below. As you will see, while the referees do highlight the interest and potential impact of your findings they also raise a number of rather major concerns that would have to be addressed before they can support publication in The EMBO Journal.

More specifically, you will see that while ref#1 is the more positive one, ref#2 is rather critical and finds that the second half of the manuscript would have to be extensively developed in order for the manuscript to be suitable for publication in The EMBO Journal. Ref #3 is largely positive, but does point out the need to provide further functional insight on the proposed role for Swi6 in limiting heterochromatin spreading.

Given the divergent recommendations from referees #1 and #2 I ran an additional round of consultation with ref#3 who provided the following comments:

' Referee 1: In general, I agree with the comments from this referee. I particularly agree with Major point 1 (a point also made myself) that it is essential to show that Swi6-EGFP is fully functional. I believe the western blot requested by referee 1 is essential for this.

Referee 2: Argues that most of the paper has little to offer over that which is already known. However, I believe the observation that expression of Swi6 from its endogenous locus (ie. not over expressed) indicates a predominantly single kinetic population of Swi6 is important as over-expression studies indicated at least two populations with different kinetics. Notwithstanding this, I do have some sympathy with his/her belief that the K9me spreading at centromeric positions in the absence of Swi6 is rather preliminary. Perhaps, the authors could enhance this section with additional data, maybe by exploring the effect of Swi6 mutants as I suggested. '

Having discussed the reports and additional comments with my colleagues in the editorial team once more, it is clear that extensive revision of your manuscript would be required to convince the referees and bring the study to the level of insight required for publication here. Given the general support from both ref #1 and #3 I would invite you to submit a revised manuscript addressing the concerns raised; however, I do also understand if you would at this stage rather prefer to seek rapid publication of the current manuscript elsewhere. In order to better define the revisions required to satisfy the referees it would be helpful if you could already at this stage provide me with an outline of the experiments that could be included in a revised manuscript. Please feel free to contact me with any questions regarding this.

In conclusion, given the referees' overall positive recommendations, I would like to invite you to submit a revised version of the manuscript, addressing the comments of all three reviewers. I should add that it is EMBO Journal policy to allow only a single round of revision, and acceptance of your manuscript will therefore depend on the completeness of your responses in this revised version. In light of the extensive, additional work required in the present case we could offer to extend the deadline for the revised manuscript.

When preparing your letter of response to the referees' comments, please bear in mind that this will form part of the Review Process File, and will therefore be available online to the community. For more details on our Transparent Editorial Process, please visit our website: http://emboj.emboress.org/about#Transparent_Process

Thank you for the opportunity to consider your work for publication. I look forward to your revision.

REFEREE COMMENTS

Referee #1:

Heterochromatin, one of the two major states of chromatin present in eukaryotic cells, was believed for decades to be static. However, live microscopy analyses together with studies on RNA-mediated heterochromatin formation have demonstrated recently that heterochromatin is a rather dynamic state. Heterochromatin Protein 1 (HP1) type proteins are chromodomain-containing proteins binding to methylated lysine 9 of histone H3 (H3K9) and are considered to be building blocks of heterochromatin. In this manuscript, entitled "H3K9 methylation extends across natural boundaries of heterochromatin in the absence of an HP1 protein", Stunnenberg and co-workers report that, in the yeast *S. pombe*, the HP1 protein Swi6 exists mostly as a single nuclear population rapidly moving on and out of heterochromatin, as well as from one site of heterochromatin to another. By using different mutant cells, it is shown that Swi6 mobility is slowed down by its binding to either methylated lysine 9 of histone H3 or RNA. Moreover, Swi6 mobility at pericentromeric and subtelomeric heterochromatin stays the same in G1, S and G2 phases of the cell cycle. Surprisingly, using H3K9 methylation chromatin immunoprecipitation (ChIP) combined to massive sequencing, it is shown that Swi6, which plays only a minor role in maintaining a high level of H3K9 methylation

at pericentric heterochromatin region, is actually required to avoid propagation of this epigenetic mark above the natural borders of this region. The requirement for Swi6 in restricting the spreading of H3K9 methylation over natural barriers is also observed at subtelomeric regions.

This is a good and an interesting study that completes and refines the conclusion of previous work on the mobility of HP1 proteins. Importantly, the specific use of a more "physiological" expression system, to avoid erroneous interpretation caused by overexpression of HP1 proteins, demonstrates that Swi6 exists only as one dynamic population of proteins when analyzed by fluorescent microscopy. This is in contrast with what was previously proposed. Importantly also, this study reports the quite unexpected finding that Swi6 HP1 protein is required for avoiding H3K9 methylation propagation over natural borders of heterochromatin.

Altogether, this study brings significant advances on the dynamics of a well-studied HP1 protein and on the requirement of this protein in constraining the propagation of H3K9 methylation, a key epigenetic mark. The findings will be of interest to a broad readership in the field of chromatin biology. I have few concerns listed below.

Major points:

1. A major strength of this work is the use of a fully functional Swi6-EGFP tagged protein expressed under Swi6 endogenous promoter. From reading the text, it is not completely clear whether the level of this protein was checked and whether it is indeed similar to the endogenous Swi6. Because, the cellular level of Swi6 protein is critical to estimate properly the dynamics of the protein, as the authors indicate, it is important to double check that Swi6-EGFP and endogenous Swi6 are expressed at similar levels within the cell. Anti-Swi6 antibodies exist and allow such control by western blot analysis. The blot should be presented in the manuscript.

2. In the line-scan FRAP experiment, the measurement done in *clr4Δ* cells does not appear to be very informative. As it is being presented (Figure 1H and main text), the analysis does not discriminate between euchromatin and heterochromatin regions. In this case, the measurement is pretty much the same as the euchromatin measurement. Moreover, it does not inform on the mobility of Swi6 in the vicinity of DNA regions undergoing heterochromatin formation in wild-type cells.

A more informative measure would be to localize such genomic regions by using specific fluorescent markers, such as CenpA for centromeres and Taz1 for telomeres, and to examine the dynamics of Swi6 in the close environment of these regions. Beyond the fact that this would be a better control, such experiment might actually show that Swi6 dynamics in these regions is in fact slower since heterochromatic RNAs accumulate strongly in *clr4Δ* cells. The available NLS-Swi6 and NLS-Swi6-KR25A constructions could also be used to further explore this possibility.

Minor points:

1. There are two HP1-like proteins in *S. pombe*, Swi6 and Chp2. Is what is found with Swi6 also true for Chp2? What is the impact of deleting *chp2* gene on H3K9 methylation propagation? Unless this is addressed, the authors should make it clear that they are referring to Swi6/HP1 both in the title and the abstract of the MS. Moreover, they should mention the existence of two HP1 proteins in *S. pombe* at least in the introduction and discussion sections of the paper.

2. The authors convincingly show that Swi6 is required for avoiding propagation of H3K9 methylation beyond natural borders of heterochromatin. However, this finding does not necessarily imply that Swi6 has a function per se in blocking such propagation. This requirement could in fact be an indirect consequence of the loss of Swi6 lead to changes in the dynamics of heterochromatin formation and maintenance, and not a direct function of the protein. This possibility should therefore be mentioned in the discussion section.

3. Swi6 deletion is having different effects on heterochromatin spreading over natural borders of heterochromatin and over a reporter gene inserted within heterochromatin. In the first case, absence of Swi6 causes a spreading of H3K9me2 whereas in the second case it leads to a loss of H3K9me over the gene. How to explain this difference and reconcile this apparent contradiction with the proposed model? Mentioning and discussing this in the discussion could be useful.

4. The model needs to be improved to make it more readable and less confusing. There are multiple aspects being presented in the scheme without necessarily a good schematic representation of these aspects. For example, the stop of heterochromatin spreading by three different systems is not easy to catch from the current scheme. Swi6⁺ and swi6^Δ situation are not represented in an equivalent manner along the green and red band. It gives the impression that in swi6^Δ cells methylation of H3K9 does not spread over a barrier, the contrary of what is shown in the manuscript.

5. Page 14, the second last sentence of the last paragraph of the result section is not clear [At certain subtelomeric regions, Swi6 appears partially redundant in ... spreading of heterochromatin]. Swi6 is redundant with what?

6. Title of Figure 4, the use of the word "heterochromatin" is an overstatement. The formulation "methylation of H3 K9" or "H3K9 me2" would be more accurate and suitable. Showing the spreading of H3K9 methylation is not sufficient to conclude that this is actually spreading of heterochromatin.

Referee #2:

In this manuscript, two parts can be distinguished. In the first part which represents the main of the paper, by FRAP analysis, the authors show that Swi6 is highly mobile and binds only transiently to constitutive heterochromatin throughout the cell cycle. Swi6 dynamics are slowed down in the presence of the histone methyltransferase Clr4 or when they expressed in the cells a Swi6 mutant unable to bind to RNA. In the second part, by ChIP-sequencing analysis of H3K9me₂, they demonstrate that Swi6 is required to prevent H3K9me₂ spreading at centromeres.

The manuscript mostly focuses on the *in vivo* dynamics of Swi6 (3 figures out of 5). They performed FRAP on endogenous tagged Swi6-EGFP and they could confirm what it is already known since more than 10 years that HP1 proteins are highly mobile in yeast and in mammals. It is also not surprising that RNA decelerated Swi6 dynamics as it is the case in the absence of Clr4. Indeed, it is expected that mobility decreases when the number of interactors (RNA or H3K9me) increases. Thus this part of the manuscript does not bring any novelty to the field. Most importantly, it is not related to the second part.

The interesting part of this manuscript is the second one putting forward a role for Swi6 in the spreading of the H3K9me₂ mark at centromeres. This certainly explains why they entitle their manuscript « H3K9 methylation extends across natural boundaries of heterochromatin in the absence of an HP1 protein », although this part corresponds only to the panels A and C of the figure 4. This observation is potentially interesting and could expand our understanding of heterochromatin spreading at centromeres. However, these data are at a preliminary stage of analysis and need mechanistic improvement before publication can be envisaged.

The manuscript in its present form is not recommended for EMBO J.

Referee #3:

Stunnenberg and colleagues have used a number of techniques, largely focussed on FRAP methodology, to investigate the dynamic kinetics of the heterochromatin protein Swi6 in *S. pombe* cells. They report that, in contrast to earlier reports, the vast majority of Swi6 exists as a single dynamic population capable of switching rapidly between different genomic locations. The main difference between these studies and the earlier ones is that here Swi6 is not over-expressed and is expressed from its endogenous locus. Interestingly, the authors also show that loss of Swi6 leads to spreading of H3K9me at specific centromeric sites. These findings are thought provoking and they will be of interest to a reasonably wide readership. The manuscript is clearly written and the experiments appear to have been performed with a high degree of technical competence. However, there are a number of issues that the authors need to address.

Much of this manuscript uses a Swi6-EGFP (or derivative) fusion protein. It is claimed on page 6 that the fusion protein is fully function and the authors cite both published work and data not shown. It is essential to substantiate this important claim with as much data as possible. Therefore, the 'data not shown' must be shown to provide full proof for the claim.

The authors have used of a hinge swap version of Swi6 that no longer binds RNA. It is a real shame that they did not also analyse other mutant versions of Swi6 in their analyses. For example, chromodomain (CD) point mutation(s) could have been included to investigate how binding of Swi6 to H3K9me affects its kinetics. I appreciate that they studied the kinetics in cells lacking Clr4 (the sole H3K9 methyltransferase in pombe) and found they increased Swi6 kinetics by approximately 50ms, however complete removal of K9me is a rather blunt assay as it disrupts other pathways and interactions. The Swi6 CD mutant would be a 'cleaner' approach in my opinion. In a similar way, Swi6 chromo-shadow domain (CSD) mutants that inhibit Swi6 dimerization could have been analysed. These may have been particularly informative with respect to the observed spreading of H3K9me at specific centromeric locations.

The increase in t1/2 values (535ms to 790ms) induced by inclusion of the NLS (Fig. 2A) is concerning as it obviously affects Swi6 kinetics via unknown and presumably non-physiological mechanisms. Therefore, data generated by this fusion protein need to be interpreted with caution.

The authors state on page 12 that their results indicate that "telomeric heterochromatin appears permissive for RNA synthesis". However, their results simply suggest that RNA may be there, but there are no data to indicate that it was transcribed locally.

1st Revision - authors' response

28 July 2015

We would like to thank all reviewers for their overall positive recommendations. We are particularly grateful to reviewer 1 and 3 for their constructive feedback, which helped us to further substantiate our findings.

Besides additional experiments proposed by the reviewers, we have performed several experiments that revealed insights into the H3K9me2 spreading mechanism that is negatively regulated by Swi6. Firstly, our new results further support a direct role of Swi6 in demarcating heterochromatin. Secondly, we show that methylation of H3K9 in the absence of Swi6 is strictly dependent on the polymerization property of Tas3, a subunit of the RNA-induced transcriptional silencing (RITS) complex. The new data added to the revised version of our manuscript reveal intriguing details about the roles of Swi6 and the RNAi pathway in maintaining centromeric heterochromatin, which will be of interest to a broad audience.

Below we will respond to the reviewers' questions/suggestions point-by-point.

Detailed response to reviewer's comments

(The referees' comments are in italics, grey. Our responses are in black.)

Referee #1:

*Heterochromatin, one of the two major states of chromatin present in eukaryotic cells, was believed for decades to be static. However, live microscopy analyses together with studies on RNA-mediated heterochromatin formation have demonstrated recently that heterochromatin is a rather dynamic state. Heterochromatin Protein 1 (HP1) type proteins are chromodomain-containing proteins binding to methylated lysine 9 of histone H3 (H3K9) and are considered to be building blocks of heterochromatin. In this manuscript, entitled "H3K9 methylation extends across natural boundaries of heterochromatin in the absence of an HP1 protein", Stunnenberg and co-workers report that, in the yeast *S. pombe*, the HP1 protein Swi6 exists mostly as a single nuclear population rapidly moving on and out of heterochromatin, as well as from one site of heterochromatin to another. By using different mutant cells, it is shown that Swi6 mobility is slowed down by its binding to either methylated lysine 9 of histone H3 or RNA. Moreover, Swi6 mobility at pericentromeric and subtelomeric heterochromatin stays the same in G1, S and G2 phases of the cell cycle. Surprisingly, using H3K9 methylation chromatin*

immunoprecipitation (ChIP) combined to massive sequencing, it is shown that Swi6, which plays only a minor role in maintaining a high level of H3K9 methylation at pericentric heterochromatin region, is actually required to avoid propagation of this epigenetic mark above the natural borders of this region. The requirement for Swi6 in restricting the spreading of H3K9 methylation over natural barriers is also observed at subtelomeric regions.

This is a good and an interesting study that completes and refines the conclusion of previous work on the mobility of HP1 proteins. Importantly, the specific use of a more "physiological" expression system, to avoid erroneous interpretation caused by overexpression of HP1 proteins, demonstrates that Swi6 exists only as one dynamic population of proteins when analyzed by fluorescent microscopy. This is in contrast with what was previously proposed. Importantly also, this study reports the quite unexpected finding that Swi6 HP1 protein is required for avoiding H3K9 methylation propagation over natural borders of heterochromatin.

Altogether, this study brings significant advances on the dynamics of a well-studied HP1 protein and on the requirement of this protein in constraining the propagation of H3K9 methylation, a key epigenetic mark. The findings will be of interest to a broad readership in the field of chromatin biology. I have few concerns listed below.

Major points:

1. A major strength of this work is the use of a fully functional Swi6-EGFP tagged protein expressed under Swi6 endogenous promoter. From reading the text, it is not completely clear whether the level of this protein was checked and whether it is indeed similar to the endogenous Swi6. Because, the cellular level of Swi6 protein is critical to estimate properly the dynamics of the protein, as the authors indicate, it is important to double check that Swi6-EGFP and endogenous Swi6 are expressed a similar levels within the cell. Anti-Swi6 antibodies exist and allow such control by western blot analysis. The blot should be presented in the manuscript.

We have included a Western Blot in Figure EV1A, which shows that Swi6-EGFP levels are similar to Swi6. We have also included a quantitative real-time RT-PCR experiment (Fig EV1B), demonstrating full functionality of Swi6-EGFP (see also below).

*2. In the line-scan FRAP experiment, the measurement done in *clr4Δ* cells does not appear to be very informative. As it is being presented (Figure 1H and main text), the analysis does not discriminate between euchromatin and heterochromatin regions. In this case, the measurement is pretty much the same as the euchromatin measurement. Moreover, it does not inform on the mobility of Swi6 in the vicinity of DNA regions undergoing heterochromatin formation in wild-type cells. A more informative measure would be to localize such genomic regions by using specific fluorescent markers, such as CenPA for centromeres and Taz1 for telomeres, and to examine the dynamics of Swi6 in the close environment of these regions. Beyond the fact that this would be a better control, such experiment might actually show that Swi6 dynamics in these regions is in fact slower since heterochromatic RNAs accumulate strongly in *clr4Δ* cells. The available NLS-Swi6 and NLS-Swi6-KR25A constructions could also be used to further explore this possibility.*

In this experiment, *clr4D* cells serve as a control. As pointed out by this reviewer, Swi6 mobility in the nucleoplasm of *wild type* cells is very similar to Swi6 mobility in *clr4D* cells. Therefore we can conclude that Swi6 is highly mobile throughout the nucleus, exchanging between different heterochromatin regions.

The additional line-scan FRAP experiments in *clr4D* cells would be technically very challenging. Anyhow, such an experiment seems redundant with experiments shown in Figure 2, which already demonstrate that Swi6 dynamics is slower when heterochromatic RNAs accumulate in *cid14D* cells. We note that RNA accumulating in *clr4D* cells can no longer be considered "heterochromatic", which is the case in *cid14D* cells. Therefore, we believe that the proposed experiments are dispensable.

Minor points:

*1. There are two HP1-like proteins in *S. pombe*, Swi6 and Chp2. Is what is found with Swi6 also true for Chp2? What is the impact of deleting *chp2* gene on H3K9 methylation propagation? Unless this is addressed, the authors should make it clear that they are referring to Swi6/HP1 both*

in the title and the abstract of the MS. Moreover, they should mention the existence of two HP1 proteins in S. pombe at least in the introduction and discussion sections of the paper.

We have amended the text to make clear that there are two HP1-like proteins in fission yeast. In addition, we have deleted *chp2+* and we have assessed H3K9me2 spreading by ChIP. In contrast to *swi6D* cells, we do not observe spreading of H3K9me2 in *chp2D* cells. This data is now shown in Figure EV5B.

2. The authors convincingly show that Swi6 is required for avoiding propagation of H3K9 methylation beyond natural borders of heterochromatin. However, this finding does not necessarily imply that Swi6 has a function per se in blocking such propagation. This requirement could in fact be an indirect consequence of the loss of Swi6 lead to changes in the dynamics of heterochromatin formation and maintenance, and not a direct function of the protein. This possibility should therefore be mentioned in the discussion section.

To address this point, we have assessed H3K9me2 spreading in Swi6 point mutants that either prevent dimerization of Swi6 (*swi6L315E*) or abolish binding to H3K9me2 (*swi6W104A*). As observed in the absence of the Swi6 proteins, also these mutants cause spreading of H3K9 methylation. These results strongly argue against indirect effects caused by the absence of the Swi6 protein. In contrast, they reveal that dimerization and H3K9me2 binding are properties of Swi6 that are crucial to avoid spreading of H3K9me2 beyond natural heterochromatin borders. The new results are shown in Figure 5.

3. Swi6 deletion is having different effects on heterochromatin spreading over natural borders of heterochromatin and over a reporter gene inserted within heterochromatin. In the first case, absence of Swi6 causes a spreading of H3K9me2 whereas in the second case it leads to a loss of H3K9me over the gene. How to explain this difference and reconcile this apparent contradiction with the proposed model? Mentioning and discussing this in the discussion could be useful.

This statement is incorrect. H3K9 methylation of a centromeric *otr::ura4+* reporter does also occur independently of Swi6 (Sadaie et al, 2004; Motamedi et al, 2008, Li et al, 2009).

4. The model needs to be improved to make it more readable and less confusing. They are multiple aspects being presented in the scheme with out necessarily a good schematic representation of these aspects. For example, the stop of heterochromatin spreading by three different systems is not easy to catch from the current scheme. Swi6+ and swi6 delta situation are not represented in an equivalent manner along the green and red band. It gives the impression that in swi6Δ cells methylation of H3K9 does not spread over a barrier, the contrary of what is show in the manuscript.

Thank you. We have changed the model figure accordingly.

5. Page 14, the second last sentence of the last paragraph of the result section is not clear [At certain subtelomeric regions, Swi6 appears partially redundant in ... spreading of heterochromatin]. Swi6 is redundant with what?

We meant to say “partially contributing to spreading” at certain subtelomeres and we have revised the text accordingly.

6. Title of Figure 4, the use of the word "heterochromatin" is an overstatement. The formulation " methylation of H3 K9" or "H3K9 me2" would be more accurate and suitable. Showing the spreading of H3K9 methylation is not sufficient to conclude that this is actually spreading of heterochromatin.

We changed the title of this figure to “Methylation of H3K9 spreads across natural boundaries and heterochromatin becomes derepressed in *swi6Δ* cells.“

Referee #2:

In this manuscript, two parts can be distinguished. In the first part which represents the main of the paper, by FRAP analysis, the authors show that Swi6 is highly mobile and binds only transiently to constitutive heterochromatin throughout the cell cycle. Swi6 dynamics are slowed down in the

presence of the histone methyltransferase Clr4 or when they expressed in the cells a Swi6 mutant unable to bind to RNA. In the second part, by ChIP-sequencing analysis of H3K9me2, they demonstrate that Swi6 is required to prevent H3K9me2 spreading at centromeres.

The manuscript mostly focuses on the in vivo dynamics of Swi6 (3 figures out of 5). They performed FRAP on endogenous tagged Swi6-EGFP and they could confirm what it is already known since more than 10 years that HP1 proteins are highly mobile in yeast and in mammals. It is also not surprising that RNA decelerated Swi6 dynamics as it is the case in the absence of Clr4. Indeed, it is expected that mobility decreases when the number of interactors (RNA or H3K9me) increases. Thus this part of the manuscript does not bring any novelty to the field. Most importantly, it is not related to the second part.

The interesting part of this manuscript is the second one putting forward a role for Swi6 in the spreading of the H3K9me2 mark at centromeres. This certainly explains why they entitle their manuscript « H3K9 methylation extends across natural boundaries of heterochromatin in the absence of an HP1 protein », although this part corresponds only to the panels A and C of the figure 4. This observation is potentially interesting and could expand our understanding of heterochromatin spreading at centromeres. However, these data are at a preliminary stage of analysis and need mechanistic improvement before publication can be envisaged. The manuscript in its present form is not recommended for EMBO J.

We disagree that the first part of our paper does not bring any novelty to the field and we are grateful to reviewers 1 and 3 for highlighting the significance of our work. It is well linked to the second part, which does now include many additional experiments that we have performed for the revised manuscript (see responses to the other reviewers).

Referee #3:

Stunnenberg and colleagues have used a number of techniques, largely focussed on FRAP methodology, to investigate the dynamic kinetics of the heterochromatin protein Swi6 in S. pombe cells. They report that, in contrast to earlier reports, the vast majority of Swi6 exists as a single dynamic population capable of switching rapidly between different genomic locations. The main difference between these studies and the earlier ones is that here Swi6 is not over-expressed and is expressed from its endogenous locus. Interestingly, the authors also show that loss of Swi6 leads to spreading of H3K9me at specific centromeric sites. These findings are thought provoking and they will be of interest to a reasonably wide readership. The manuscript is clearly written and the experiments appear to have been performed with a high degree of technical competence. However, there are a number of issues that the authors need to address.

Much of this manuscript uses a Swi6-EGFP (or derivative) fusion protein. It is claimed on page 6 that the fusion protein is fully function and the authors cite both published work and data not shown. It is essential to substantiate this important claim with as much data as possible. Therefore, the 'data not shown' must be shown to provide full proof for the claim.

Full functionality of Swi6-EGFP is now shown in Figure EV1B. We have also included a Western Blot, which shows that Swi6-EGFP levels are similar to Swi6 (Figure EV1A).

The authors have used of a hinge swap version of Swi6 that no longer binds RNA. It is a real shame that they did not also analyse other mutant versions of Swi6 in their analyses. For example, chromodomain (CD) point mutation(s) could have been included to investigate how binding of Swi6 to H3K9me affects its kinetics. I appreciate that they studied the kinetics in cells lacking Clr4 (the sole H3K9 methyltransferase in pombe) and found they increased Swi6 kinetics by approximately 50ms, however complete removal of K9me is a rather blunt assay as it disrupts other pathways and interactions. The Swi6 CD mutant would be a 'cleaner' approach in my opinion. In a similar way, Swi6 chromo-shadow domain (CSD) mutants that inhibit Swi6 dimerization could have been analysed. These may have been particularly informative with respect to the observed spreading of H3K9me at specific centromeric locations.

It has been previously demonstrated that Swi6 requires both the CSD and the CD to display full binding activity in vivo (Cheutin et al, 2004). Therefore, we don't think that repeating such

measurements for CD and CSD mutants by FRAP would add much to the paper. However, we highly appreciate the suggestion to assess the spreading of H3K9me2 in these Swi6 mutants, which provided further support for the model we are proposing. The new data is shown in Figure 5 of the revised manuscript (see also response to reviewer 1).

The increase in t1/2 values (535ms to 790ms) induced by inclusion of the NLS (Fig. 2A) is concerning as it obviously affects Swi6 kinetics via unknown and presumably non-physiological mechanisms. Therefore, data generated by this fusion protein need to be interpreted with caution.

The results obtained in conjunction with these constructs reveal that RNA production within heterochromatin is greatly fluctuating and stochastic throughout the cell cycle. Because the NLS constructs are not used to make claims about the exact binding kinetics of Swi6, we are not concerned.

The authors state on page 12 that their results indicate that "telomeric heterochromatin appears permissive for RNA synthesis". However, their results simply suggest that RNA may be there, but there are no data to indicate that it was transcribed locally.

RNA transcripts are being synthesized from telomeric heterochromatin. We have included the raw data of a quantitative RT-PCR experiment in the revised version of the manuscript that demonstrates this clearly (Figure EV1C).

Additional round of consultation with Referee 3 with regard to comments from Referee 2

' Referee 1: In general, I agree with the comments from this referee. I particularly agree with Major point 1 (a point also made myself) that it is essential to show that Swi6-EGFP is fully functional. I believe the western blot requested by referee 1 is essential for this.

Swi6-EGFP is fully functional and expressed to similar levels as Swi6. This additional data is shown in Figure EV1 of the revised manuscript. We note that we can only fit Swi6-EGFP kinetics to a two-component model if we express Swi6-EGFP from a plasmid using the same condition as in Cheutin et al. (0.4µg/ml thiamine). Even though EGFP might slightly stabilize Swi6, this has no consequence on spreading or silencing of heterochromatin and we are not able to detect a static fraction by FRAP. We therefore conclude that this fusion protein is fully functional.

Referee 2: Argues that most of the paper has little to offer over that which is already known. However, I believe the observation that expression of Swi6 from its endogenous locus (ie. not over expressed) indicates a predominantly single kinetic population of Swi6 is important as over-expression studies indicated at least two populations with different kinetics. Notwithstanding this, I do have some sympathy with his/her belief that the K9me spreading at centromeric positions in the absence of Swi6 is rather preliminary. Perhaps, the authors could enhance this section with additional data, maybe by exploring the effect of Swi6 mutants as I suggested. '

As acknowledged by all three reviewers, our finding that H3K9me2 spreads across natural boundaries in the absence of Swi6 is highly unexpected and thought provoking. Therefore, we believe that publication of this result on its own would have been fully justified. However, we are able to relate to the request for more mechanistic insight and we have therefore added a substantial amount of new data to the revised manuscript.

As already alluded to above, we have performed additional experiments with various Swi6 point mutants. The new results that we show in Figure 5 of the revised manuscript further support a direct role of Swi6 in counteracting the spreading of H3K9 methylation and allow us to propose a model in which Swi6 dimers, when adopting a specific conformation (Canzio et al, 2013), prevent the spreading of H3K9 methylation. We find this possibility intriguing and it will be exciting to explore this further.

In addition, we have performed experiments to test the hypothesis that spreading in the absence of Swi6 is mediated by self-association of the RITS subunit Tas3. Our hypothesis is based on a paper by the Moazed and Patel labs (Li et al, Molecular Cell, 2009), which demonstrated that a C-terminal Tas3 alpha-helical motif (TAM) promotes Tas3 self-association and thereby cis-spreading of the RITS complex. Indeed, we found that the ability of Tas3 to interact with itself is absolutely required

for the methylation of H3K9me2 if Swi6 is missing.

2nd Editorial Decision

23 August 2015

Thank you for submitting a revised version of your manuscript. It has now been seen by two of the original referees whose comments are shown below.

As you will see they both find that all criticisms have been sufficiently addressed and recommend the manuscript for publication. However, before we can proceed to officially accept your manuscript there are a few editorial issues concerning text and figures that I would ask you to address:

-> Please rephrase the legend and text describing fig 6 as suggested by ref #1

-> Please fill out and include an author checklist as listed in our online guidelines (<http://emboj.embopress.org/authorguide>)

Thank you again for giving us the chance to consider your manuscript for The EMBO Journal, I look forward to your final revision.

 REFEREE COMMENTS

Referee #1:

In this revised MS, Stunnenberg and co-workers provide a series of controls and new findings that further support the main conclusions made in the original version. These findings also provide additional insights into the role of a HP1 protein in restricting the spreading of H3K9 methylation over natural barriers of heterochromatin.

More specifically, the authors convincingly addressed the following points and issues, which were raised in my comments of the original manuscript.

First, the level and functionality of Swi6-GFP (expressed from *swi6* locus) was checked and compared to wild-type Swi6. Western blot and RT-qPCR results show no pronounced difference in the levels of Swi6-GFP protein and mRNA, compared to wild-type Swi6. A measure of the efficiency of silencing in cells expressing Swi6-GFP (in place of Swi6) also shows that these cells have a heterochromatin gene silencing similar to wild-type cells.

Second, the implication of Chp2, the second HP1-type protein present in *S. pombe*, in regulating the spreading of H3K9 methylation was tested. The results indicate that only Swi6 is required for limiting such spreading.

Third, the possibility that the observed propagation of H3K9 methylation is an indirect consequence of *swi6* deletion was addressed. Analysis of cells expressing Swi6 point mutants, which are defective for either Swi6 homodimerization or binding to H3K9me2, shows that they have similar extended propagation of H3K9 methylation. These results further support the idea that Swi6 is directly implicated in limiting H3K9me propagation.

Following on experiments asked or suggested by the other reviewers, the authors also provide more findings on what protein is responsible for the propagation of H3K9 methylation in absence of Swi6. It is found that this propagation is mediated by the protein Tas3 and requires Tas3's capacity to homodimerize.

Altogether, the additional experiments conducted and the new findings added in the revised manuscript significantly improve the quality and significance of this study.

I have only a minor comment on the revised version. Analysis of Tas3 requirement in the spreading of H3K9 methylation is limited to a pericentromeric region and not addressed for other heterochromatin regions. This should be indicated in figure 6 legend and the main text.

Referee #3:

The authors have done a good job in answering my prior concerns. They have included additional data which strengthens the manuscript by providing further mechanistic insight.